



**Effect of soil saturation on denitrification in a grassland soil**
Laura Maritza Cardenas[1*], Roland Bol, R.[2], Dominika Lewicka-Szczebak,[3], Andrew Stuart
Gregory[4], Graham Peter Matthews[5], William Richard Whalley[4], Thomas Henry Misselbrook[1],
David Scholefield[1] and Reinhard Well[3]
[1]Rothamsted Research, North Wyke, Okehampton, Devon EX20 2SB, United Kingdom
[2]Institute of Bio- and Geosciences, IBG-3/Agrosphere, Forschungszentrum Jülich GmbH, 52428
Jülich, Germany
[3]Thünen Institute of Climate-Smart Agriculture, Federal Research Institute for Rural Areas,
Forestry and Fisheries, Bundesallee, 50, D-38116 Braunschweig, Germany
[4]Rothamsted Research, Harpenden, Hertfordshire AL5 2JQ, United Kingdom
[5]University of Plymouth, Drake Circus, Plymouth, Devon PL4 8AA, United Kingdom
*Correspondence to: Laura M. Cardenas (laura.cardenas@rothamsted.ac.uk)
**Abstract.** Nitrous oxide ($N_2O$) is of major importance as a greenhouse gas and precursor of
ozone ($O_3$) destruction in the stratosphere mostly produced in soils. The soil emitted $N_2O$ is
predominantly derived from denitrification and to a smaller extent, nitrification in soils, both
processes controlled by environmental factors and their interactions, and are influenced by
agricultural management. Soil water content expressed as water filled pore space (WFPS) is a major
controlling factor of emissions and its interaction with compaction, has not been studied at the
micropore scale. A laboratory incubation was carried out at different saturation levels for a
grassland soil and emissions of $N_2O$ and $N_2$ were measured as well as the isotopomers of $N_2O$. We
found that fluxes variability was larger in the less saturated soils probably due to nutrient
distribution heterogeneity created from soil cracks and consequently nutrient hot spots. The results
agreed with denitrification as the main source of fluxes at the highest saturations, but nitrification
could have occurred at the lower saturation, even though moisture was still high (71% WFSP). The
isotopomer data showed isotopic similarities in the wettest treatments vs the two drier ones; and





results agreed with previous findings where it is clear there are 2 N-pools with different dynamics:
added N producing intense denitrification, vs soil N resulting in less isotopic fractionation.
**Keywords**
Grassland, nitrous oxide, isotopomers, isotopologues, greenhouse gases

## 1 Introduction

Nitrous oxide ($N_2O$) is of major importance as a greenhouse gas and precursor of ozone ($O_3$)
destruction in the stratosphere (Crutzen, 1970). Agriculture is a major source of greenhouse gases
(GHGs), such as carbon dioxide ($CO_2$), methane ($CH_4$) and also $N_2O$ (IPCC, 2006). The application
of organic and inorganic fertiliser N to agricultural soils enhances the production of $N_2O$ (Baggs *et*
*al.*, 2000). This soil emitted $N_2O$ is predominantly derived from denitrification and to a smaller extent,
nitrification in soils (Davidson and Verchot, 2000). Denitrification is a microbial process in which
reduction of nitrate ($NO_3^-$) occurs to produce $N_2O$, and $N_2$ is the final product of this process, benign
for the environment, but represents a loss of N in agricultural systems. Nitrification is an oxidative
process in which ammonium ($NH_4^+$) is converted to $NO_3^-$ (Davidson and Verchot, 2000). Both
processes are controlled by environmental factors and their interactions, and are influenced by
agricultural management (Firestone and Davidson, 1989). It is well recognised that soil water content
expressed as water filled pore space (WFPS) is a major controlling factor and as Davidson (1991)
illustrated, nitrification is a source of $N_2O$ until WFPS values reach about 70%, after which
denitrification dominates. In fact, Firestone and Davidson (1989) gave oxygen supply a ranking of 1
in importance as a controlling factor in fertilised soils, above C and N. At WFPS between 45 and
75% a mixture of nitrification and denitrification act as $N_2O$ sources. Davidson also suggested that at
WFPS values above 90% only $N_2$ is produced. Several studies have later proposed models to relate
WFPS with emissions (Schmidt *et al.*, 2000; Dobbie and Smith, 2001; Parton *et al.*, 2001; del Prado
*et al.*, 2006; Castellano *et al.*, 2010) but the "optimum" WFPS for $N_2O$ emissions varies from soil to
soil (Davidson, 1991). Soil structure could be influencing this effect and it has been identified to
strongly interact with soil moisture (Ball *et al.*, 1999; van Groenigen *et al.*, 2005) through changes in



WFPS. Particularly soil compaction due to livestock treading and the use of heavy machinery affect
soil structure and emissions as shown by studies relating bulk density to fluxes (Klefoth *et al.*, 2014b);
and degrees of tillage to emissions (Ludwig *et al.*, 2011).

Compaction is known to affect the size of the larger pores (macropores) thereby reducing the

soil air volume and therefore increasing the WFPS (for the same moisture content) (van der Weerden
*et al.*, 2012). However, little is known about the effect of compaction on the smaller soil pores
(micropores) and this could provide valuable information for understanding the simultaneous
behaviour of the dynamics of water in the various pore sizes in soil. Such an understanding would
lead to the development of better $N_2O$ mitigation strategies via dealing with soil compaction issues.

The role of water in soils is closely linked to microbial activity but also relates to the degree

of aeration and gas diffusivity in soils (Morley and Baggs, 2010). Water facilitates nutrient supply to
microbes and restricts gas diffusion, thereby increasing the residence time of gases in soil, and the
chance of further $N_2O$ reduction before it can be released to the atmosphere. This is further aided by
the restriction of the diffusion of atmospheric $O_2$ (Dobbie and Smith, 2001), increasing the potential
for denitrification. As a consequence, counteracting effects (high microbial activity vs low diffusion)
occur simultaneously making it difficult to predict net processes and corresponding outputs
(Davidson, 1991). Detailed understanding of the sources of $N_2O$ and the influence of physical factors,
i.e. soil structure and its interaction with moisture, is a powerful tool for developing effective
mitigation strategies.

Isotopologues of $N_2O$ represent the isotopic substitution of the O and/or the two N atoms

within the $N_2O$ molecule. The isotopomers of $N_2O$, are those differing in the peripheral ($\beta$) and central
N-positions ($\alpha$) of the linear molecule (Toyoda and Yoshida, 1999) with the intramolecular $^{15}N$ site
preference (SP; the difference between $\delta^{15}N^{\alpha}$ - $\delta^{15}N^{\beta}$) used to identify production processes at the
level of microbial species or enzymes involved (Toyoda *et al.*, 2005; Ostrom, 2011). Moreover, $\delta^{18}O$,
$\delta^{15}N$ and SP of emitted $N_2O$ depend on the denitrification product ratio ($N_2O$ / ($N_2$+$N_2O$)), and hence
provide insight into the dynamics of $N_2O$ reduction (Well and Flessa, 2009; Lewicka-Szczebak *et al.*,



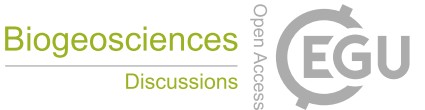

2015; Lewicka-Szczebak *et al.*, 2014). Data reported in the literature provide values for these
parameters in relation to the source process for N$_2$O. Koster *et al*. (2013) for example recently
reported $\delta^{15}N^{bulk}$ values of N$_2$O between –36.8‰ and –31.9‰ in the conditions of their experiment,
which are indicative of denitrification according to Perez *et al.* (2006) and Well and Flessa (2009)
who proposed the range –54 to –10‰ relative to the substrate. Baggs (2008) summarised that values
between –90 to –40‰ are indicative of nitrification. Determination of these values are normally
carried out in pure culture studies or in conditions favouring either production or reduction of N$_2$O
(Well and Flessa, 2009). The SP is however considered a better predictor of the N$_2$O source due to
its independence from the substrate signature (Ostrom, 2011).

Simultaneous occurrence production and reduction of N$_2$O as in natural conditions presents

a challenge for isotopic factors determination due to uncertainty on N$_2$ reduction and the co-existence
of different microbial communities resulting in other steps of denitrification happening as well
(Lewicka-Szczebak *et al.*, 2014). Recently, using data from the experiment here reported, where soil
was incubated under aerobic atmosphere and the complete denitrification process occurs, Lewicka-
Szczebak *et al.* (2015) determined fractionation factors associated with N$_2$O production and reduction
using a modelling approach. The analysis comprised measurements of the N$_2$O and N$_2$ fluxes
combined with isotopomer data. The results generally confirmed the range of values of η (net isotope
effects) and η$^{18}$O/η$^{15}$N ratios reported by previous studies for N$_2$O reduction for the soil volume
reached by the N+C amendment. This did not apply for the soil volume not reached by the N+C
amendment.

Lewicka-Szczebak *et al.* (2015) observed a clear relationship between $^{15}$N and $^{18}$O isotope

effects during N$_2$O production and denitrification rates. For N$_2$O reduction, differential isotope effects
were observed for two distinct soil pools characterized by different product ratios N$_2$O / (N$_2$+N$_2$O).
For moderate product ratios (from 0.1 to 1.0) the range of isotope effects given by previous studies
was confirmed and refined, whereas for very low product ratios (below 0.1) the net isotope effects
were much smaller. In this paper, we present the results from the gas emissions measurements from





soils collected from a long-term permanent grassland soil to assess the impact of different levels of
soil saturation on $N_2O$ and $N_2$ and $CO_2$ emissions after compaction. The measurements included the
soil isotopomer ($^{15}N_\alpha$, $^{15}N_\beta$ and site preference) analysis of emitted $N_2O$, which in combination with
the bulk $^{15}N$ and $^{18}O$ was used to distinguish between $N_2O$ from bacterial denitrification and other
processes (e.g. nitrification and fungal denitrification) (Lewicka-Szczebak, 2016a).

The study was carried out under laboratory controlled conditions, using a specialised

laboratory denitrification (DENIS) incubation system (Cardenas *et al.*, 2003). The system allows
continuous measurements of N gases as well as $CO_2$, and spot sampling for isotopomer analyses.

We conducted measurements at defined saturation of pores size fractions as a prerequisite to

model denitrification as a function of water status (Butterbach Bahl *et al.*, 2013 and Müller and
Clough, 2014). We have under controlled conditions created a single compaction stress of 200 kPa
in incremental layers using a uniaxial pneumatic piston to simulate a grazing pressure. We
hypothesized that at high water saturation, heterogeneity in N emissions decreases due to more
homogeneous distribution of the soil nutrients and/or anaerobic microsites. We also hypothesized that
even at high soil moisture a mixture of nitrification and denitrification can occur. We aimed to
understand changes in the ratio $N_2O/(N_2O+N_2)$ and the behaviour and utility of isotopologues of $N_2O$
at the different moisture levels studied in a controlled manner on soil micro and macropores.
**2 Materials and methods**
**2.1 Soil used in the study**
An agricultural soil, under grassland management since at least 1838 (Barré *et al.*, 2010), was
collected from a location adjacent to a long-term ley-arable experiment at Rothamsted Research in
Hertfordshire (Highfield, see soil properties in Table 1 and further details in Rothamsted Research,
2006; Gregory *et al.*, 2010). The soil had been under permanent cut mixed-species (predominantly
*Lolium* and *Trifolium*) vegetation. The soil was sampled as described in Gregory *et al.* (2010). Briefly
it was sampled from the upper 150 mm of the profile, air dried in the laboratory, crumbled and sieved



(<4 mm), mixed to make a bulk sample and equilibrated at a pre-determined water content (37 g 100
g$^{-1}$; Gregory *et al.*, 2010) in air-tight containers at 4° C for at least 48 hours.
**1.2. Preparation of soil blocks**
The equilibrated soil was then packed into twelve stainless steel blocks (145 mm diameter; h: 100
mm), each of which contained three cylindrical holes (i.d: 50 mm; h: 100 mm each), to a single
compaction stress of 200 kPa in incremental layers using a uniaxial pneumatic piston. The three hole-
blocks were used to facilitate the compression of the cores. The 200 kPa stress was analogous to a
severe compaction event by a tractor (Gregory *et al.*, 2010) or livestock (Scholefield *et al.*, 1985).
The total area of the upper surface of soil in each block was therefore 58.9 cm$^2$ (3 × 19.6 cm$^2$) and
the target volume of soil was set to be 544.28 cm$^3$ (3 × 181.43 cm$^3$) with the objective of leaving a
headspace of approximately 45 cm$^3$ (3 × 15 cm$^3$) for the subsequent experiment. The precise height
of the soil (and hence the volume) was measured using the displacement measurement system of a
DN10 Test Frame (Davenport-Nene, Wigston, Leicester, UK) with a precision of 0.001 mm.
**2.3 Equilibration of soil cores at different saturations**
The soil was equilibrated to four different initial saturation conditions or treatments (t0) which were
based on the likely distribution of water between macropores and micropores. The first treatment was
where both the macro- and micropores (and hence the total soil) was fully saturated; the second
treatment was where the macropores were half-saturated and the micropores remained fully saturated;
the third treatment was where the macropores were fully unsaturated and the micropores again
remained fully saturated; and the fourth treatment was where the macropores were fully unsaturated
and the micropores were half-saturated. These four treatments are hereafter referred to as SAT/sat;
HALFSAT/sat; UNSAT/sat and UNSAT/halfsat, respectively, where upper-case refers to the
saturation condition of the macropores and lower-case refers to the saturation condition of the
micropores. In order to set these initial saturation conditions, we referred to the gravimetric soil water
release characteristic for the soil, as given in Gregory *et al.* (2010), which represents the assumed





pore size distribution, and a fitted van Genuchten function (van Genuchten, 1980) with the Mualem
(1976) constraint of m = 1−1/n:
$$\theta_h = \theta_r + \frac{\theta_s - \theta_r}{[1 + (\alpha h)^n]^{1 - \frac{1}{n}}}$$     [1]

where $\theta_s$, $\theta_r$ and $\theta_h$ are the saturated, residual (water content at permanent wilting point) and

$h$ matric potential gravimetric water contents (g g$^{-1}$), respectively; $h$ is the matric potential (|kPa|, i.e.
the absolute value), $\alpha$ is a fitted parameter approximating the inverse of $h$ at the inflection point (|kPa|$^-$
$^1$), often linked to the air-entry point, and m and n are dimensionless fitted parameters related to the
shape of the function.

The somewhat arbitrary saturation state known as "field capacity" represents the idealised

condition UNSAT/sat, where the macropores have drained and the micropores have yet to drain. As
field capacity has typically corresponded to a matric potential anywhere between -5 to -33 kPa, we
chose -20 kPa as our UNSAT/sat condition, where the threshold pores size between water-filled pores
at this matric potential is 15 µm. The matric potential corresponding to SAT/sat was obviously 0 kPa,
to give full saturation of all the pores. To calculate the intermediate HALFSAT/sat condition, we took
the mid-point gravimetric water content between 0 and -20 kPa from the water release characteristic,
and calculated the corresponding matric potential using Eq. [1], which was -8.6 kPa. We also
calculated the mid-point gravimetric water content between that at -20 kPa and $\theta_r$, and found the
corresponding matric potential (Eq. [1]) to be -78.1 kPa. We used this to represent the UNSAT/halfsat
condition. As $\theta_r$ was non-zero (in fact it was 0.236 g g$^{-1}$), due to the fine-textured nature of the soil,
we accept that at -78.1 kPa the micropores were not truly half-saturated but would have been in a
wetter condition than this. However due to our method for equilibrating the soils prior to
experimentation, we required a suitable matric potential not lower than -1500 kPa that we could
control in the laboratory (see below). It could be argued that trying to attain a water content in the
hygroscopic range (that held at potentials much lower than -1500 kPa, often in the vapour phase),
where the true mid-point water content between that at -20 kPa and complete dryness in this soil lay,





was not especially relevant to denitrification processes expected in such a soil. There was one final
adjustment to make. The subsequent incubation experiment was to involve a 15 ml ($3 \times 5$ ml) addition
of solution (see below). Through knowing masses and volumes of the solid-water-air phases of our
blocks, we therefore calculated revised matric potentials which would mean that the subsequent
addition of solution would achieve the target potentials given above. The target matric potentials of
0 (SAT/sat), -8.6 (HALFSAT/sat), -20.0 (UNSAT/sat) and -78.1 kPa (UNSAT/halfsat) were revised
to -4.1, -12.3, -27.3 and -136.9 kPa, respectively (see summary in Table 2). For the SAT/sat and
HALFSAT/sat conditions, two sets of three replicate blocks were placed on two fine-grade sand
tension tables connected to a water reservoir. For the UNSAT/sat condition a set of three replicate
blocks was placed on a tension plate connected to a water reservoir, and the final set of three replicate
blocks were placed in pressure plate chambers connected to high-pressure air. All blocks were
saturated on their respective apparatus for 24 h, and were then equilibrated for 7 days at the adjusted
target matric potentials which were achieved by either lowering the water level in the reservoir (sand
tables and tension plate) or by increasing the air pressure (pressure chambers). At the end of
equilibration period, the blocks were removed carefully from the apparatus, wrapped in air-tight film,
and maintained at 4 °C until the subsequent incubation.
**2.4 Incubation**
Each block containing three cores was placed in an individual incubation vessel of the automated
laboratory system described by Cardenas *et al.* (2003) in a randomised block design to avoid effect
of vessel. The lids for the vessels containing three holes were lined with the cores in the block to
ensure that the solution to be applied later would fall on top of each soil core. Stainless steel bulkheads
fitted (size for ¼" tubing) on the lids had a three-layered Teflon coated silicone septum (4 mm thick
x 7 mm diameter) for supplying the amendment solution by using a gas tight hypodermic syringe.
The bulkheads were covered with a stainless steel nut and only open when amendment was applied.
The incubation experiment lasted 13 days. The incubation vessels with the soils were contained in a
temperature controlled cabinet and the temperature set at 20°C. The incubation vessels were flushed





from the bottom at a rate of 30 ml min$^{-1}$ with a He/O$_2$ mixture (21% O$_2$, natural atmospheric
concentration) for 24 h, or until the system and the soils atmosphere were emitting low background
levels of both N$_2$ and N$_2$O (N$_2$ can get down to levels of 280 ppm much smaller than atmospheric
values). Subsequently, the He/O$_2$ supply was reduced to 10 ml min$^{-1}$ and directed across the soil
surface and measurements of N$_2$O and N$_2$ carried out at approximately 2 hourly cycles to sample from
all the 12 vessels. Emissions of CO$_2$ were simultaneously measured.

**2.5 Application of amendment**

An amendment solution equivalent to 75 kg N ha$^{-1}$ and 400 kg C ha$^{-1}$ was applied as a 5 ml aliquot a
solution containing KNO$_3$ and glucose to each of the three cores in each vessel on day 0 of the
incubation. Glucose is added to optimise conditions for denitrification to occur (Morley and Baggs,
2010). The aliquot was placed in a stainless steel container (volume 1.2 l) which had three holes
drilled with bulkheads fitted, two to connect stainless steel tubing for flushing the vessel, and the third
one to place a septum on a bulkhead to withdraw solution. Flushing was carried out with He for half
an hour before the solution was required for application to the soil cores and continued during the
application process to avoid atmospheric N$_2$ contamination (a total of one and a half hours). The
amendment solution was manually withdrawn from the container with a glass syringe fitted with a
three-way valve onto the soil surface; care was taken to minimise contamination from atmospheric
N$_2$ entering the system. The syringe content was injected to the soil cores via the inlets on the lids
consecutively in each lid (three cores) and all vessels, completing a total of 36 applications that lasted
about 45 minutes. Incubation continued for twelve days, and the evolution of N$_2$O, N$_2$ and CO$_2$
measured continuously. At the end of each incubation experiment, the soils were removed from the
incubation vessels for further analysis. The three cores in each incubation vessel were pulled together
in one sample and subsamples taken and analysed for mineral N, total N and C and moisture status.
Table 3 shows the results of the soil analysis for all cores.





**2.6 Gas measurements**
Gas samples were directed to the relevant analysers via an automated injection valve fitted with 2
loops to direct the sample to two gas chromatographs. Emissions of $N_2O$ and $CO_2$ were measured by
Gas Chromatography (GC), fitted with an Electron Capture Detector (ECD) and separation achieved
by a stainless steel packed column (2 m long, 4 mm bore) filled with 'Porapak Q' (80–100 mesh) and
using $N_2$ as the carrier gas. The detection limit for $N_2O$ was equivalent to 2.3 g N ha$^{-1}$ d$^{-1}$. The $N_2$ was
measured by GC with a He Ionisation Detection (HID) and separation achieved by a PLOT column
(30 m long 0.53 mm i.d.), with He as the carrier gas. The detection limit was 9.6 g N ha$^{-1}$ d$^{-1}$. The
response of the two GCs was assessed by measuring a range of concentrations for $N_2O$, $CO_2$ and $N_2$.
Parent standards of the mixtures 10133 ppm $N_2O$ + 1015.8 ppm $N_2$; 501 ppm $N_2O$ + 253 ppm $N_2$ and
49.5 ppm $N_2O$ + 100.6 ppm $N_2$ were diluted by means of Mass Flow controllers with He to give a
range of concentrations of: for $N_2O$ of up to 750 ppm and for $N_2$ 1015 ppm. For $CO_2$ a parent standard
of 30,100 ppm was diluted down to 1136 ppm (all standards were in He as the balance gas). Daily
calibrations were carried out for $N_2O$ and $N_2$ by using the low standard and doing repeated
measurements. The temperature inside the refrigeration cabinet containing the incubation vessels was
logged on an hourly basis and checked at the end of the incubation. The gas outflow rates were also
measured and recorded daily, and subsequently used to calculate the flux.
**2.7 Measurement of $N_2O$ isotopic signatures**
Gas samples for isotopologue analysis were collected in 115 ml serum bottles sealed with grey butyl
crimp-cap septa (Part No 611012, Altmann, Holzkirchen, Germany). The bottles were connected by
a Teflon tube to the end of the chamber vents and were vented to the atmosphere through a needle, to
maintain flow through the experimental system. Dual isotope and isotopomer signatures of $N_2O$, i.e.
$\delta^{18}O$ of $N_2O$ ($\delta^{18}O-N_2O$), average $\delta^{15}N$ ($\delta^{15}N^{bulk}$) and $\delta^{15}N$ from the central N-position ($\delta^{15}N^\alpha$) were
analysed after cryo-focussing by isotope ratio mass spectrometry as described previously (Well *et al.*,
2008). $^{15}N$ site preference (SP) was obtained as SP = 2 * ($\delta^{15}N^\alpha$ – $\delta^{15}N^{bulk}$). Dual isotope and
isotopomer ratios of a sample ($R_{sample}$) were expressed as ‰ deviation from $^{15}N/^{14}N$ and $^{18}O/^{16}O$ ratios





of the reference standard materials ($R_{std}$), atmospheric $N_2$ and standard mean ocean water (SMOW),
respectively:

$$\delta X = (R_{sample}/R_{std} - 1) \times 1000 \qquad [2]$$

where $X = {}^{15}N^{bulk}, {}^{15}N^{\alpha}, {}^{15}N^{\beta}, or {}^{18}O$

**263    2.8 Data analysis and additional measurements undertaken**

The areas under the curves for the $N_2O$, $CO_2$ and $N_2$ data were calculated by using GenStat 11 (VSN
International Ltd, Hemel Hempstead, Herts, UK). The resulting areas for the different treatments were
analysed by applying analysis of variance (*ANOVA*). The isotopic (${}^{15}N^{bulk}$, ${}^{18}O$, and site preference
(SP) differences between the four treatment for the different sampling dates were analysed by two-
way ANOVA. We also used the Student's *t* test to check for changes in soil water content over the
course of the experiments.

Calculation of the relative contribution of the $N_2O$ derived from bacterial denitrification

(%$B_{DEN}$) was done according to Lewicka-Szczebak *et al.* (2015). The isotopic value of initially
produced $N_2O$, *i.e.* prior to its partial reduction ($\delta_0$) was determined using a Rayleigh model (Mariotti
*et al.*, 1982), were $\delta_0$ is calculated using the fractionation factor of $N_2O$ reduction ($\eta_{N2O-N2}$) for SP and
the fraction of residual $N_2O$ ($r_{N2O}$) which is equal to the $N_2O/(N_2+N_2O)$ product ratio obtained from
direct measurements of $N_2$ and $N_2O$ flux. An endmember mixing model is then used to calculate the
percentage of bacterial $N_2O$ in the total $N_2O$ flux (%$B_{DEN}$) from calculated $\delta_0$ values and the SP and
$\delta^{18}O$ endmember values of bacterial denitrification and fungal denitrification/nitrification. The range
in endmember and $\eta_{N2O-N2}$ values assumed (adopted from Lewicka-Szczebak, 2016a) to calculated
maximum and minimum estimates of %$B_{DEN}$ is given in Table 4.

Because both, endmember values and $\eta_{N2O-N2}$ values are not constant but subject to the given

ranges, we calculated here several scenarios using combinations of maximum, minimum and average
endmember and $\eta_{N2O-N2}$ values (Table 4) to illustrate the possible range of %$B_{DEN}$ for each sample.
For occasional cases where %$B_{DEN} > 100\%$ the values were set to 100%.





At the same time as preparing the main soil blocks, a set of replicate samples was prepared in
exactly the same manner, but in smaller cores (i.d: 50 mm; h: 25 mm). On these samples we analysed
soil mineral N, total N and C and moisture at the start of the incubation. The same parameters were
measured after incubation by doing destructive sampling from the cores. Mineral N ($NO_3^-$, $NO_2^-$ and
$NH_4^+$) was analysed after extraction with KCl by means of a segmented flow analyser using a
colorimetric technique (Searle, 1984). Total C and N in the air dried soil were analysed using a
thermal conductivity detector (TCD, Carlo Erba, model NA2000). Soil moisture was determined by
gravimetric analysis after drying at 105°C.
**3 Results**
**3.1 Soil composition**
The results after moisture adjustment at the start of the experiment resulted in a range of WFPS of
100 to 71% for the 4 treatments (Table 2). The results from the end of the incubation also showed
that there remained significant differences in soil moisture between the high moisture treatments
(SAT/sat and HALFSAT/sat) and the two lower moisture treatments (Table 3; one-way ANOVA,
$p<0.05$). Soil in the two wettest states lost statistically significant amounts of water (10% ($p=0.006$)
and 4.4% ($p<0.001$) for SAT/sat and HALFSAT/sat, respectively) over the course of the 13-day
incubation experiment. This was inevitable as there was no way to hold a high (near-saturation) matric
potential once the soil was inside the DENIS assembly, and water would have begun to drain by
gravitational forces out of the largest macropores (>30 µm). An additional factor was the continuous
$He/O_2$ delivery over the soil surface which would have caused some drying. We accepted these as
unavoidable features of the experimental set-up, but we suggest that the main response of the gaseous
emissions occurred under the initial conditions, prior to the loss of water over subsequent days. Soil
in the two drier conditions had no significant change in their water content over the experimental
period ($p= 0.153$ and $0.051$ for UNSAT/sat and UNSAT/halfsat, respectively). The results of the
initial soil composition were, for mineral N: 85.5 mg $NO_3^-$-N kg$^{-1}$ dry soil, 136.2 mg $NH_4^+$-N kg$^{-1}$ dry
soil. The mineral N contents of the soils at the end of the incubation are reported in Table 3 showing





that $NO_3^-$ was very small in treatments SAT/sat and HALFSAT/sat (~1 mg N $kg^{-1}$ dry soil) compared
to UNSAT/sat and UNSAT/halfsat (50-100 N $kg^{-1}$ dry soil) at the end of the incubation. Therefore,
there was a significant difference in soil $NO_3^-$ between the former, high moisture treatments and the
latter drier (UNSAT) treatments which were also significantly different between themselves ($p<0.001$
for both). The $NH_4^+$ content was similar in treatments SAT/sat, HALFSAT/sat and UNSAT/sat (~100
mg N $kg^{-1}$ dry soil), but slightly lower in treatment UNSAT/halfsat (71.3 mg N $kg^{-1}$ dry soil), however
overall differences were not significant ($p>0.05$).
**3.2 Gaseous emissions of $N_2O$, $CO_2$ and $N_2$**
The results showed that for treatments SAT/sat and HALFSAT/sat all three gases, $N_2O$, $CO_2$ and $N_2$
showed fluxes that were well replicated for all the vessels (see Fig. 1), in contrast for UNSAT/sat and
UNSAT/halfsat the emissions between the various replicated vessel in each treatment was not as
consistent, leading to a larger within treatment variability in the magnitude and shape of the GHG
fluxes measured. The cumulative fluxes also resulted in larger variability for the drier treatments
(Table 3).
*Nitrous oxide and nitrogen gas*. The general trend was that the $N_2O$ concentrations in the
headspace increased shortly after the application of the amendment (Fig. 1). The duration of the $N_2O$
peak for each replicate soil samples was about three days, except for UNSAT/halfsat in which one of
the replicate soils exhibit a peak which lasted for about 5 days. The $N_2O$ maximum in the SAT/sat
and HALFSAT/sat treatments was of similar magnitude (ca. 5.5 kg N $ha^{-1}$ $d^{-1}$) and those of
UNSAT/sat and UNSAT/halfsat also were comparable (at around 7 kg N $ha^{-1}$ $d^{-1}$). The $N_2$
concentrations always increased before the soil emitted $N_2O$ reached the maximum. The lag between
both $N_2O$ and $N_2$ peak for all samples was only few hours. Peaks of $N_2$ generally lasted just over four
days, except in UNSAT/halfsat where one replicate lasted about 6 days (Fig. 1). Unlike in the $N_2O$
data, there was larger within treatment variability in the replicates for all four treatments. The standard
deviations of each mean (Table 3) also indicate the large variability in treatments UNSAT/sat and
UNSAT/halfsat for both $N_2O$ and $N_2$.



The product ratios, i.e. $N_2O/(N_2O+N_2)$ showed a peak just after amendment addition by ca.
0.73 (at 0.49 d), 0.65 (at 0.48 d), 0.99 (at 0.35 d) and 0.88 (at 0.42 d) for SAT/sat, HALFSAT/sat,
UNSAT/sat and UNSAT/halfsat, respectively, and then decreases gradually until day 3 where it
becomes nearly zero for the 2 wettest treatments, and stays stable for the driest treatments between
0.1-0.2 (see Table 5 showing the daily means of these ratios).
The cumulative areas of the $N_2O$ and $N_2$ peaks analysed by one-way ANOVA resulted in no
significant differences between treatments for both $N_2O$ and $N_2$ (Table 3). Due to the large variation
in treatments UNSAT/sat and UNSAT/halfsat we carried out a pair wise analysis by using a weighted
t-test (Cochran, 1957). This analysis showed treatment differences between SAT/sat and
HALFSAT/sat, HALFSAT/sat and UNSAT/sat, SAT/sat and UNSAT/sat, but only at the 10%
significance level (P <0.1 for both $N_2O$ and $N_2$). It is possible that gases were trapped (particularly in
the higher saturation treatments) due to low diffusion and thus possibly masked differences in $N_2$ and
$N_2O$ production since this fraction of gases was not detected (Harter et al. 2016).
The results showed that the total N emission $(N_2O+N_2)$ (Table 3) had a consistent decreasing
trend, with decreasing soil moisture i.e. from 63.4 for SAT/sat (100% WFPS) to 34.1 kg N ha$^{-1}$ (71%
WFPS) for UNSAT/halfsat. The maximum cumulative $N_2O$ occurred at around 80% WFPS as Fig. 2
shows. The total $N_2O+N_2$ was largest at about 95% and for $N_2$ it was our upper treatment at 100%
WFPS.
*Carbon dioxide*. The background $CO_2$ values (before amendment application, i.e. day -1 to
day 0) were high at around 30 kg C ha$^{-1}$ d$^{-1}$ and variable (not shown). The $CO_2$ concentrations in the
headspace increased within a few hours after amendment application. The maximum $CO_2$ flux was
reached earlier in the drier treatments (about 1-2 days; ~70 kg C ha$^{-1}$ d$^{-1}$) compared to the wettest (3
days; ~40 kg C ha$^{-1}$ d$^{-1}$) and former peaks were also sharper (Fig. 1). The cumulative $CO_2$ fluxes were
significantly larger in the two drier unsaturated treatments (ca. 400-420 kg C ha$^{-1}$) when compared to
the wetter more saturated treatment (ca. 280-290 kg C ha$^{-1}$, P<0.05) (Table 3).





### 3.3 Isotopologues of N$_2$O

The $\delta^{15}N^{bulk}$ of the soil emitted N$_2$O in our study differed significantly among the four treatments and between the seven sampling dates (p<0.001 for both); there was also a significant treatment*sampling date interaction (p<0.001). The maximum $\delta^{15}N^{bulk}$ generally occurred on day 3, except for SAT/sat on day 4 (Table 6).

The maximum $\delta^{18}O$-N$_2$O values were also found on day 3, except for SAT/sat which peaked at day 2 (Table 6). Overall, the $\delta^{18}O$-N$_2$O values varied significantly between treatment and sampling dates (p<0.001 for both), but there was no significant treatment*time interaction (p>0.05).

The site preference (SP) showed for the SAT/sat treatment an initial maximum value on day 2 (6.3‰) which decreased thereafter in period from day 3 to 5 to a mean SP values of the emitted N$_2$O of 2.0‰ on day 5, subsequently rising to 8.4‰ on day 12 of the experiment (Table 6). The HALFSAT/sat treatment had the highest initial SP values on day 2 and 3 (both 6.4‰), decreasing again to a value of 2.0‰, but now already on day 4 followed by subsequent higher SP values of up to 9.2‰ on day 7 (Table 6). The two driest treatments (UNSAT/sat and UNSAT/halfsat) both showed an initial maximum on day 3 (11.9‰ and 5.9‰, respectively), and in UNSAT/sat the SP value then decreased to day 7 (3.9‰), but in UNSAT/halfsat treatment after a marginal decrease on day 4 (5.4‰) it then increased throughout the experiment reaching 11.8‰ on day 12 (Table 6). The lowest SP values were generally on day 1 in all treatments. Overall, for all parameters, there was more similarity between the more saturated treatments SAT/sat and HALFSAT/sat, and between the two more dry and aerobic treatments UNSAT/sat and UNSAT/halfsat.

The plot of the N$_2$O / (N$_2$O + N$_2$) ratio vs SP for all treatments in the first two days (when N$_2$O was increasing and the N$_2$O / (N$_2$O + N$_2$) ratio was decreasing) shows a significant negative response of the SP when the ratio increased (Fig. 3). The regression suggests that when the emitted gaseous N is dominated by N$_2$O (ratio close to 1) the SP values will be slightly negative with values around -2 (Fig. 3). This is in juxtaposition with the situation when the N emissions are dominated by N$_2$ or N$_2$O is low, where the SP values of soil emitted N$_2$O were much higher (Fig. 3), pointing to an



overall product ratio related to an 'isotopic shift' of 10 to 12.5‰. We fitted 3 functions through this
data including a second degree polynomial, a linear and logarithmic function. The fitted logarithmic
function, shown in Fig. 3, is in almost perfect agreement with Lewicka-Szczebak *et al.* (2014).
Lewicka-Szczebak *et al.* (2014) data fits on the top left of Fig. 3 (their values are for SP and ratio
$N_2O / (N_2O + N_2)$: 18.5, 0.18; 10.1, 0.19; 11, 0.28 and 13.4, 0.24, respectively).
It has been reported that the combination of the isotopic signatures of $N_2O$ potentially
identifies the contribution of processes other than bacterial denitrification (Köster *et al*., 2015; Wu
Di *et al.*, 2016; Deppe *et al*., 2017) so we have carried out similar analysis with our data. The
maximum $\delta^{18}O$ and SP values, were generally observed at or near the peak of $N_2$ emissions on days
2-3, independent of the moisture treatment (Table 6 and Fig. 3). $\delta^{15}N^{bulk}$ values of all treatments were
mostly negative when $N_2O$ fluxes started to increase (day 1, Fig. 1, Table 6), except for
UNSAT/halfsat in which the lowest value was before amendment application, reaching their highest
values between days 3 and 4 for when $N_2O$ fluxes were back to the low initial values, and then
decreased during the remaining period. $\delta^{18}O$ values increased about 10 - 20‰ after day 1 reaching
maximum values on days 2 or 3 in all treatments, while SP increased in parallel, at least by 3‰
(SAT/sat) and up to 12‰(UNSAT/sat). While $\delta^{18}O$ exhibited a steady decreasing trend after day 3,
SP behaved opposite to $\delta^{15}N^{bulk}$ with decreasing values while $\delta^{15}N^{bulk}$ was rising again after days 4 or

404 5.

We further explored the data by looking at the relationships between the $\delta^{18}O$ and $\delta^{15}N^{bulk}$ for
all the treatments. Figure 4 shows the $\delta^{18}O$ vs $\delta^{15}N^{bulk}$ for all treatments separating the data in three
periods: '-1', with $\delta^{18}O$ vs $\delta^{15}N^{bulk}$ values 1 day prior to the moisture adjustment (and N and C
application); '1-2', with values in the first 2 days after the addition of water, N and C were added and
$N_2O$ emissions were generally increasing in all treatments; and, '3-12', the period in days after
moisture adjustment and N and C addition when $N_2O$ emissions generally decreased back to baseline
soil emissions. There was a strong relationship between $\delta^{18}O$ vs $\delta^{15}N^{bulk}$ for the high moisture
treatments ($R^2$= 0.973 and 0.923 for SAT/sat and HALFSAT/sat, respectively) at the beginning of





the incubation ('1-2') when the $N_2O$ emissions are still increasing, in contrast to those of the lower
soil moisture treatments that were lower ($R^2$= 0.294 and 0.622, for UNSAT/sat and UNSAT/halfsat,
respectively). The relationships between $\delta^{18}O$ vs $\delta^{15}N^{bulk}$ of emitted $N_2O$ for the '3-12' period have
$R^2$ values between 0.549 and 0.896 (Fig. 4). Interestingly, with decreasing soil moisture content (Fig.
4a to 4d) the regression lines of '1-2' and '3-12' day period got closer together in the plotted graphs.
Overall, the $\delta^{15}N^{bulk}$ isotopic distances between the two lines was larger for a given $\delta^{18}O$-$N_2O$ value
for SAT/sat and HALFSAT/sat (ca. 20‰) when compared to the UNSAT/sat and UNSAT/halfsat
treatments (ca. 13‰) (Fig. 4). So it seems the $\delta^{15}N^{bulk}$ / $\delta^{18}O$-$N_2O$ signatures are more similar for the
drier soil than the two wettest treatments. In addition, Fig 4 exactly reflects the 2-pool dynamics with
increasing $\delta^{15}N$ and $\delta^{18}O$ while the product ratio goes down (days 2,3), then only $\delta^{15}N$ continue
increasing due to fractionation of the $NO_3^-$ during exhaustion of pool 1 in the wet soil (days 3,4,5),
finally as pool 1 is depleted and more and more comes from pool 2, the product ratio increases
somewhat, and $\delta^{15}N$ decreases somewhat since pool 2 is less fractionated and also $\delta^{18}O$ decreases due
to slightly increasing product ratio. Note that the turning points of $\delta^{18}O$ and product ratio (Table 3
and 4) for the wetter soils almost coincide.
Similarly to Fig. 4, we plotted the $\delta^{18}O$ vs the SP (Fig. 5) for the different phases of the
experiment. Generally, the slopes (Table 7) for days 1-2 for the three wettest treatments were similar
(~0.2-0.3) following the range of known reduction slopes and also had high regression coefficients
($R^2$= 0.65, 0.90 and 0.87 for SAT/sat, HALFSAT/Sat and UNSAT/sat, respectively). The slopes on
days 3-5 were variable but slightly similar on days 7-12 (between 41 and 0.68) for the same three
treatments. Figure 5 also shows the "map" for the values of SP and $\delta^{18}O$ from all treatments.
Reduction lines (vectors) represent minimum and maximum routes of isotopologue values with
increasing $N_2O$ reduction to $N_2$ based on the reported range in the ratio between the isotope
fractionation factors of $N_2O$ reduction for SP and $\delta^{18}O$ (Lewicka-Szczebak et al., (2016a) Most
samples are located within the vectors (from Lewicka-Szczebak *et al.* 2016a) area of $N_2O$ production
by bacterial denitrification with partial $N_2O$ reduction to $N_2$ (within uppermost and lowermost $N_2O$



reduction vectors representing the extreme values for the bacterial endmember and reduction slopes).
Only a few values of the UNSAT/sat and UNSAT/halfsat treatments are located above that area and
more close or within the area of mixing between bacterial denitrification and fungal
denitrification/nitrification.

The estimated ranges of the proportion of emitted $N_2O$ resulting from bacterial denitrification

(%$B_{DEN}$) were on day 1 and 2 after the amendment comparable in all four moisture treatments (Table
6). However, during day 3 to 12 the %$B_{DEN}$ ranged from 78-100% in SAT/sat and 79-100%
HALFSAT/Sat, which was generally higher than that estimated at 54-86% for UNSAT/halfsat
treatment. The %$B_{DEN}$ of the UNSAT/halfsat in that period was intermediate between SAT/sat and
UNSAT/sat with range of range 60-100% (Table 6). The final values were similar to those on day -1
except for the UNSAT/sat treatment.
**4 Discussion**
**4.1 $N_2O$ and $N_2$ fluxes**
The observed decrease in total N emissions with decreasing soil moisture reflects the effect of soil
moisture as reported in previous studies (Well *et al.*, 2006). The differences when comparing the
cumulative fluxes however, were only marginally ($p<0.1$) significant (Table 3) mostly due to large
variability within replicates in the drier treatments (see Fig. 1b). Davidson *et al.* (1991) provided a
WFPS threshold for determination of source process, with a value of 60% WFPS as the borderline
between nitrification and denitrification as source processes for $N_2O$ production. The WFPS in all
treatments in our study was larger than 70%, above this 60% threshold, and referred to as the
"optimum water content" for $N_2O$ by Scheer *et al.* (2009), so we can be confident that denitrification
was likely to have been the main source process in our experiment. In addition, Bateman *et al.* (2004)
observed the largest $N_2O$ fluxes at 70% WFPS on a silty loam soil, lower than the 80% value for the
largest fluxes from the clay soil in our study (Fig. 2) suggesting that this optimum value could change
with soil type. Further, the maximum total measured N lost ($N_2O+N_2$) in our study occurred at about
95% WFPS (Fig. 2), but not many studies report $N_2$ fluxes for comparison and we are still missing



measurements of nitric oxide (NO) (Davidson *et al.*, 2000) and ammonia ($NH_3$) to account for the
total N losses. It is however possible that the $N_2O+N_2$ fluxes in the SAT/sat treatment were
underestimated due to low diffusivity in the water filled pores (Well *et al.*, 2001).
The smaller standard errors in both $N_2O$ and $N_2$ data for the larger soil moisture levels (Table
3 and Fig. 1) could suggest that at high moisture contents nutrient distribution (N and C) on the top
of the core is more homogeneous making replicate cores to behave similarly. At the lower soil
moisture for both $N_2O$ and $N_2$, it is possible that some cracks appear on the soil surface causing
downwards nutrient movement, resulting in heterogeneity in nutrient distribution on the surface and
increasing variability between replicates, reflected in the larger standard errors of the fluxes. Laudone
*et al.* (2011) studied, using a biophysical model, the positioning of the hot-spot zones away from the
critical percolation path (described as 'where air first breaks through the structure as water is removed
at increasing tensions') and found it slowed the increase and decline in emission of $CO_2$, $N_2O$ and $N_2$.
They found that hot-spot zones further away from the critical percolation path would reach the
anaerobic conditions required for denitrification in shorter time, the products of the denitrification
reactions take longer to migrate from the hot-spot zones to the critical percolation path and to reach
the surface of the system. The model and its parameters can be used for modelling the effect of soil
compaction and saturation on the emission of $N_2O$. They suggest that having determined biophysical
parameters influencing $N_2O$ production, it remains to determine whether soil structure, or simply
saturation, is the determining factor when the biological parameters are constrained. Furthermore,
Clough *et al.* (2013) indicate that microbial scale models need to be included on larger models linking
microbial processes and nutrient cycling in order to consider spatial and temporal variation. Kulkarni
*et al.* (2008) refers to "hot spots" and "hot moments" of denitrification as scale dependant and
highlight the limitations for extrapolating fluxes to larger scales due to these inherent variabilities.
Well *et al.* (2003) found that under saturated conditions there was good agreement between laboratory
and field measurements of denitrification, and attributed deviations, under unsaturated conditions, to
spatial variability of anaerobic microsites and redox potential. Dealing with spatial variability when





measuring $N_2O$ fluxes in the field remains a challenge, but the uncertainty could be potentially
reduced if water distribution is known. Our laboratory study suggests that soil $N_2O$ and $N_2$ emission
for higher moisture levels would be less variable than for drier soils and suggests that for the former
a smaller number of spatially defined samples will be needed to get an accurate field estimate.

Our results, for the two highest water contents (SAT/sat and HALFSAT/sat), showed that $N_2O$

only contributed 20% of the total N emissions, as compared to 40-50% at the lowest water contents
(UNSAT/sat and UNSAT/halfsat, Table 3). This was due to reduction to $N_2$ at the high moisture level,
confirmed by the larger $N_2$ fluxes, favoured by low gas diffusion which increased the $N_2O$ residence
time and the chance of further transformation (Klefoth *et al.*, 2014a). We should also consider the
potential underestimation of the fluxes in the highest saturation treatment due to restricted diffusion
in the water filled pores (Well *et al.*, 2001). A total of 99% of the soil $NO_3^-$ was consumed in the two
high water treatments, whereas in the drier UNSAT/sat and UNSAT/halfsat treatments there still was
35% and 70% of the initial amount of $NO_3^-$ left in the soil, at the end of the incubation, respectively
(Table 3). The total amount of gas lost compared to the $NO_3^-$ consumed was almost 3 times for the
wetter treatments, and less than twice for the 2 drier ones. This agrees with denitrification as the
dominant process source for $N_2O$ with larger consumption of $NO_3^-$ at the higher moisture and larger
$N_2$ to $N_2O$ ratios (5.7, 4.7 for SAT/sat and HALFSAT/sat, respectively), whereas at the lower
moisture, ratios were lower (1.5 and 1.0 for UNSAT/sat and UNSAT/halfsat, respectively) (Davidson,
1991). This also indicates that with WFPS above the 60% threshold for $N_2O$ production from
denitrification, there was an increasing proportion of anaerobic microsites with increase in saturation
controlling $NO_3^-$ consumption and $N_2/N_2O$ ratios in an almost linear manner. With WFPS values
between 71-100 % and $N_2/N_2O$ between 1.0 and 5.7, a regression can be estimated: $Y=0.1723 X -$
$11.82$ ($R^2=0.8585$), where Y is $N_2/N_2O$ and X is %WFPS. In summary, we propose that
heterogeneous distribution of anaerobic microsites could have been the limiting factor for complete
depletion of $NO_3^-$ and conversion to $N_2O$ in the two drier treatments. In addition, in the
UNSAT/halfsat treatment there was a decrease in soil $NH_4^+$ at the end of the incubation (almost 50%;





Table 3) suggesting nitrification could have been occurring at this water content which also agrees
with the increase in $NO_3^-$, even though WFPS was relatively high (>71%) (Table 3). It is important
to note that as we did not assess gross nitrification, the observed net nitrification based on lowering
in $NH_4^+$ could underestimate gross nitrification since there might have been substantial N
mineralisation during the incubation. However, under conditions favouring denitrification at high soil
moisture the typical $N_2O$ produced from nitrification is much lower compared to that from
denitrification (Lewicka-Szczebak *et al.*, 2016a) with the maximum reported values for the $N_2O$ yield
of nitrification of 1-3 % (e.g. Deppe *et al*., 2017). If this is the case, nitrification fluxes could not have
exceeded 1 kg N with $NH_4^+$ loss of < 30 kg * 3% ~1 kg N. This would have represented for the driest
treatment, if conditions were suitable only for one day, that nitrification-derived $N_2O$ would have
been 6% of the total $N_2O$ produced. Loss of $NH_3$ was not probable at such low pH (5.6). The
corresponding rate of $NO_3^-$ production using the initial and final soil contents and assuming other
processes were less important in magnitude, would have been < 1 mg $NO_3^-$-N kg dry soil$^{-1}$ d$^{-1}$ which
is a reasonable rate (Hatch *et al.*, 2002). The other three treatments lost similar amounts of soil $NH_4^+$
during the incubation (23-26%) which could have been due to some degree of nitrification at the start
of the incubation before $O_2$ was depleted in the soil microsites or due to $NH_4^+$ immobilisation (Table
3) (Geisseler *et al.*, 2010).
The $CO_2$ released in all treatments supports the statement above in relation with the more
aerobic status of UNSAT/sat and UNSAT/halfsat, because the cumulative $CO_2$ flux is roughly 1.5
times higher in the two drier treatments when compared to the wetter ones; but it could have also
been the result of higher diffusion in the drier treatments.
A mass N balance, taking into account the initial and final soil $NO_3^-$, $NH_4^+$, added $NO_3^-$ and
the emitted N (as $N_2O$ and $N_2$) results in unaccounted N-loss of 177.2, 177.6, 130.6 and 110.8 mg N
kg$^{-1}$ for SAT/sat, HALFSAT/sat, UNSAT/sat and UNSAT/halfsat, respectively, that could have been
emitted as other N gases (such as NO), and some, immobilised in the microbial biomass. In addition,



in the SAT/sat treatment there was probably an underestimation of the produced $N_2$ and $N_2O$ due to
restricted diffusion at the high WFPS (e.g. Well *et al.*, 2001).

**4.2 Isotopologue trends.**

Trends of isotopologue values of emitted $N_2O$ coincided with those of $N_2$ and $N_2O$ fluxes. The results
from the isotopomer data (Table 6 and Fig. 3) also showed that generally there were more isotopic
similarities between the two wettest treatments when compared to the two contrasting drier soil
moisture treatments.
Isotopologue values of emitted $N_2O$ reflect multiple processes where all signatures are
affected by the admixture of several microbial processes, the extent of $N_2O$ reduction to $N_2$ as well
as the variability of the associated isotope effects (Lewicka-Szczebak *et al.*, 2015). Moreover, for
$\delta^{18}O$ and $\delta^{15}N^{bulk}$ the precursor signatures are variable (Decock and Six, 2013), for $\delta^{18}O$ the O
exchange with water can be also variable (Lewicka-Szczebak *et al.*, 2016b). Since the number of
influencing factors clearly exceeds the number of isotopologue values, unequivocal results can only
be obtained if certain processes can be excluded or be determined independently, (Lewicka-Szczebak
*et al.*, 2015; Lewicka-Szczebak, 2016a). The two latter conditions were fulfilled in this study, i.e.
$N_2O$ fluxes were high and several order of magnitude above possible nitrification fluxes, since the
$N_2O$ – to- $NO_3^-$ ratio yield of nitrification products rarely exceeds 1% (Well *et al.*, 2008; Zhu et al.,
2012). Moreover, $N_2$ fluxes and thus $N_2O$ reduction rates were exactly quantified.
The estimated values of % $B_{DEN}$ showed that in the period immediately after amendment
application all moisture treatments were similar, reflecting that the microbial response to N and C
added was the same and denitrification dominated. This was the same for the rest of the period for
the wetter treatments. In the drier treatments, proportions decreased afterwards and were similar to
values before amendment application, possibly due to recovery of more aerobic conditions that could
have encouraged other processes to contribute. As $N_2$ was still produced in the driest treatment, (but
in smaller amounts), this indicated ongoing denitrifying conditions and thus large contributions from
nitrification were not probable, but some occurred as suggested by $NH_4^+$ consumption.



The trends observed reflect the dynamics resulting from the simultaneous application of

$NO_3^-$ and labile C (glucose) on the soil surface as described in previous studies (Meijide *et al.*,
2010; Bergstermann *et al.*, 2011) where the same soil was used, resulting in two locally distinct
$NO_3^-$ pools with differing denitrification dynamics. In the soil volume reached by the $NO_3^-$/glucose
amendment, denitrification was initially intense with high $N_2$ and $N_2O$ fluxes and rapid isotopic
enrichment of the $NO_3^-$-N. When the $NO_3^-$ and/or glucose of this first pool were exhausted, $N_2$ and
$N_2O$ fluxes were much lower and dominated by the initial $NO_3^-$ pool that was not reached by the
glucose/$NO_3^-$ amendment and that is less fractionated due to its lower exhaustion by denitrification,
causing decreasing trends in $\delta^{15}N^{bulk}$ of emitted $N_2O$.

This is also reflected in Fig 4 showing that $N_2O$ fluxes from both pools exhibited correlations

between $\delta^{15}N^{bulk}$ and $\delta^{18}O$ due to varying $N_2O$ reduction, but $\delta^{15}N^{bulk}$ values in days 1 and 2 - i.e. the
phase when Pool 1 dominated - were distinct from the previous and later phase.

The fit of $^{15}N^{bulk}$ /$^{18}O$ data to two distinct and distant regression lines can be attributed to

two facts: Firstly, in the wet treatment (Fig 4a, b) Pool 1 was probably completely exhausted and
there was little $NO_3^-$ formation from nitrification (indicated by final $NO_3^-$ values close to 0, Table 3)
whereas the drier treatment exhibited substantial $NO_3^-$ formation and high residual $NO_3^-$. Hence,
there was probably still some $N_2O$ from Pool 1 after day 2 in the dry treatment but not in the wetter
ones. Secondly, the product ratios after day 2 of the drier treatments were higher (0.13 to 0.44)
compared to the wetter treatments (0.001 to 0.09). Thus the isotope effect of $N_2O$ reduction was
smaller in the drier treatments, leading to a smaller upshift of $\delta^{15}N^{bulk}$ and thus more negative values
after day 2, i.e. with values closer to days 1 +2.

This finding further confirms that $\delta^{15}N/\delta^{18}O$ patterns are useful to identify the presence of

several N pools, e.g. typically occurring after application of liquid organic fertilizers which has
been previously demonstrated using isotopologue patterns (Koster *et al.*, 2015).

Interestingly, the highest $\delta^{15}N^{bulk}$ and $\delta^{18}O$ values of the emitted $N_2O$ were found in the soils

of the HALFSAT/sat treatment, although it may have been expected that the highest isotope values





from the N$_2$O would be found in the wettest soil (SAT/sat) because N$_2$O reduction to N$_2$ is favoured
under water-saturated conditions due to extended residence time of produced N$_2$O (Well et al., 2012).
However, N$_2$O/(N$_2$+N$_2$O) ratios of the SAT/sat and SAT/halfsat treatments were not different (Table
5). Bol *et al.* (2004) also found that some estuarine soils under flooded conditions (akin to our
SAT/sat) showed some strong simultaneous depletions (rather than enrichments) of the emitted N$_2$O
$\delta^{15}$N$^{bulk}$ and $\delta^{18}$O values. These authors suggested that this observation may have resulted from a flux
contribution of an 'isotopically' unidentified N$_2$O production pathway. Another explanation could be
complete consumption of some of the produced N$_2$O in isolated micro-niches in the SAT/sat treatment
due to inhibited diffusivity in the fully saturated pores space. N$_2$ formation in these isolated domains
would not affect the isotopologue values of emitted N$_2$O and this would thus result in lower apparent
isotope effects of N$_2$O reduction in water saturated environments as suggested by Well *et al.* (2012).

The SP values obtained were generally below 12‰ in agreement with reported ranges

attributed to bacterial denitrification: -2.5 to 1.8‰ (Sutka *et al.*, 2006); 3.1 to 8.9‰ (Well and
Flessa, 2009); -12.5 to 17.6‰ (Ostrom, 2011).  The SP, believed to be a better predictor of the N$_2$O
source as it is independent of the substrate isotopic signature (Ostrom, 2011), has been suggested as
it can be used to estimate N$_2$O reduction to N$_2$ in cases when bacterial denitrification can be
assumed to dominate N$_2$O fluxes (Koster *et al.*, 2013; Lewicka-Szczebak *et al.*, 2015). There was a
strong correlation between the SP and N$_2$O / (N$_2$O+N$_2$) ratios on the first 2 days of the incubation
for all treatments up until the N$_2$O reached its maximum (Fig. 3) which reflects the accumulation of
$\delta^{15}$N at the alpha position during ongoing N$_2$O reduction to N$_2$. Later on in the experiment beyond
day 3, this was not observed probably because in that period the product ratio remained almost
unchanged and very low (Table 6). Similar observations have been reported by Meijide *et al.* (2010)
and Bergstermann *et al.* (2011), as they also found a decrease in SP during the peak flux period in
total N$_2$+N$_2$O emissions, but only when the soil had been kept wet prior to the start of the
experiment (Bergstermann *et al.*, 2011). These results confirm from 2 independent studies
(Lewicka-Szczebak *et al.*, 2014) that there is a relationship between the product ratios and isotopic





signatures of the N₂O emitted. The $\delta^{18}O$ vs SP regressions showed more similarity between the
three wettest treatments as well as high regression coefficients, suggesting this SP/$\delta^{18}O$ ratio could
also be used to help identify patterns for emissions and their sources.
**4.3 Link to modelling approaches.**
Since isotopologue data could be compared to N₂ and N₂O fluxes, the variability of isotope effects
of N₂O production and reduction to N₂ by denitrification could be determined from this data set
(Lewicka-Szczebak *et al.*, 2015). This included modelling the two pool dynamics discussed above.
It was shown that net isotope effects of N₂O reduction ($\eta_{N2O-N2}$) determined for both NO₃⁻ pools
differed. Pool 1 representing amended soil and showing high fluxes but moderate product ratio,
exhibited $\eta_{N2O-N2}$ values and the characteristic $\eta^{18}O/\eta^{15}N$ ratios similar to those previously reported,
whereas for Pool 2 characterized by lower fluxes and very low product ratio, the net isotope effects
were much smaller and the $\eta^{18}O/\eta^{15}N$ ratios, previously accepted as typical for N₂O reduction
processes (i.e., higher than 2), were not valid.

The question arises, if the poor coincidence of Pool 2 isotopologue fluxes with previous

N₂O reduction studies reflects the variability of isotope effects of N₂O reduction or if the
contribution of other processes like fungal denitrification could explain this.

Liu *et al.* (2016) noted that on the catchment scale potential N₂O emission rates were

related to hydroxylamine and NO₃⁻, but not NH₄⁺ content in soil. Zou *et al.* (2014) found high SP
(15.0 to 20.1‰) values at WFPS of 73 to 89% suggesting that fungal denitrification and bacterial
nitrification contributed to N₂O production to a degree equivalent to that of bacterial denitrification.

To verify the contribution of fungal denitrification and/or hydroxylamine oxidation we can

first look at the $\eta SP_{N2O-NO3}$ values calculated in the previous modelling study applied on the same
dataset, (Table 1, the final modelling Step, Lewicka-Szczebak *et al.*, 2015). For Pool 1 there are no
significant differences between the values of various treatments, $SP_0$ ranges from (-1.8±4.9) to
(+0.1±2.5). Pool 1 emission was mostly active in days 1-2, hence these values confirm the bacterial
dominance in the emission at the beginning of incubation, which originates mainly from the



amendment addition and represent similar pathway for all treatments. However, for the Pool 2
emission we could observe a significant difference when compared the two wet treatments (SAT/sat
and HALFSAT/sat: (-5.6±7.0)) with the UNSAT/sat treatment (+3.8±5.8). This represents the
emission from unamended soil which was dominating after the third day of the incubation and
indicates higher nitrification contribution for the drier treatment.
**4.4 Contribution of bacterial denitrification.**
An endmember mixing approach has been previously used to estimate the fraction of bacterial $N_2O$
(%$B_{DEN}$), but without independent estimates of $N_2O$ reduction (Zou *et al.*, 2014), but due to the
unknown isotopic shift by $N_2O$ reduction, the ranges of minimum and maximum estimates were large,
showing that limited information is obtained without $N_2$ flux measurement.

In an incubation study with two arable soils, Koster *et al.* (2013) used $N_2O/(N_2+N_2O)$ ratios

and isotopologue values of gaseous fluxes to calculate SP of $N_2O$ production (referred to as $SP_0$),
which is equivalent to $SP_0$ using the Rayleigh model and published values of $\eta_{N2O-N2}$. The
endmember mixing approach based on $SP_0$ was then used to estimate fungal denitrification and/or
hydroxylamine oxidation giving indications for a substantial contribution in a clay soil, but not in a
loamy soil. Here we presented for the first time an extensive data set with large range in product
ratios and moisture to calculate the contribution of bacterial denitrification (%$B_{DEN}$) of emitted $N_2O$
from $SP_0$. The uncertainty of this approach arises from three factors, (i) from the range of $SP_0$
endmember values for bacterial denitrification of -11 to 0 per mil and 30 to 37 for hydroxylamine
oxidation/fungal denitrification, (ii) from the range of net isotope effect values of $N_2O$ reduction
($\eta_{N2O-N2}$) for SP which vary from -2 to -8 per mil (Lewicka-Szczebak *et al.*, 2015), and iii) system
condition (open vs. closed) taken to estimate the net isotope effect (Wu *et al.*, 2016).

The observation that %$B_{DEN}$ of emitted $N_2O$ was generally high (63-100%) in the wettest

treatment (SAT/sat) was not unexpected. However interestingly %$B_{DEN}$ in the HALFSAT/sat
treatment was very similar (71-98%), pointing to the role of the wetter areas of the soil
microaggregates contributing to high %$B_{DEN}$ values. The slightly lower values, i.e. down 60% in



UNSAT/sat $\%B_{DEN}$ range of 60-100%, suggest that the majority of $N_2O$ derived from bacterial
denitrification still results from the wetter microaggregates of the soils, despite the fact that the
macropores are now more aerobic. Only, when the micropores become partially wet, as in the
UNSAT/halfsat treatment, do the more aerobic soil conditions allow a higher contribution of
nitrification/fungal denitrification ranging from 0 - 46% (1 - % $B_{DEN}$, Table 6) on days 3-12 (Zhu *et*
*al.*, 2013). Differences in the contribution of nitrification/fungal denitrification between the flux
phases when different $NO_3^-$ pools were presumably dominating are only indicated in the driest
treatment, since 1-%$B_{DEN}$ was higher after day 2 (14 to 46%) compared to days 1+2 (0 to 33 %).
This larger share of nitrification/fungal denitrification can be attributed to the increasing
contribution from Pool 2 to the total flux as indicated by the modeling of higher $SP_0$ for Pool 2 (see
previous section and Lewicka-Szczebak *et al.* (2015). In addition, indication for elevated
contribution of processes other than bacterial denitrification were only evident in the drier
treatments during phases before and after $N_2$, $N_2O$ fluxes were strongly enhanced by glucose
amendment. The data supply no clue whether the other processes were suppressed during the anoxia
induced by glucose decomposition or just masked by the vast glucose-induced bacterial $N_2O$ fluxes.

## 5 Conclusions

The results from this study showed that at high soil moisture levels, there was less variability in N
fluxes between replicates, potentially decreasing the importance of soil hot spots in emissions at
these moisture levels. At high moisture there also was complete depletion of nitrate confirming
denitrification as the main pathway for $N_2O$ emissions, and due to less diffusion of the produced
$N_2O$, the potential for further reduction to $N_2$ increased. Under less saturation, but still relatively
high soil moisture, nitrification occurred. Isotopic similarities were observed between similar
saturation levels and patterns of $\delta^{15}N/\delta^{18}O$ and $SP/\delta^{18}O$ are suggested as indicators of source
processes.



**Acknowledgments**
The authors would like to thank the technical help from Mark Butler during the laboratory
incubation and Andrew Bristow and Patricia Butler for carrying out soil analysis. Also thanks to
Dan Dhanoa for advice on statistical analysis, and to Anette Giesemann and Martina Heuer for help
in $N_2O$ isotopic analyses. This study was funded by the UK Biotechnology and Biological Sciences
Research Council (BBSRC) with competitive grants BB/E001580/1 and BB/E001793/1.
Rothamsted Research is sponsored by the BBSRC.




**Figures**

**Figure 1**. Mean of the three replicates for $N_2O$, $N_2$ and $CO_2$ emissions from a. SAT/sat treatment; b. HALFSAT/sat; c. UNSAT/sat; d. UNSAT/halfsat. Grey lines correspond to the standard error of the means.

**Figure 2** Total N emissions ($N_2O+N_2$)-N, $N_2O$ and $N_2$ vs WFPS. Fitted functions through each dataset are also shown.

**Figure 3** Ratio $N_2O$ / ($N_2O + N_2$) vs. Site Preference (SP) for all for treatments in the first two days. A logarithmic function was fitted through the data, the corresponding equation and correlation coefficient are given.

**Figure 4** $\delta^{18}O$ vs $\delta^{15}N_{bulk}$ in all treatments for three periods (day -1 in diamond symbol, days 1-2 in square symbol and days 3-12 in triangle symbol, respectively) in the experiment: a. SAT/sat treatment; b. HALFSAT/sat; c. UNSAT/sat; d. UNSAT/halfsat. Equations of fitted functions and correlation coefficients are shown.

**Figure 5** Site Preference vs $\delta^{18}O$ in all treatments for three periods (day -1, days 1-2 and days 3-12) in the experiment: a. SAT/sat treatment; b. HALFSAT/sat; c. UNSAT/sat; d. UNSAT/halfsat. Equations of fitted functions and correlation coefficients are in Table 7 for 1-2, 3-5 and 7-12 (5-12 for c.). Endmember areas for nitrification, N; bacterial denitrification, D; fungal denitrification, FD and nitrifier denitrification, ND and corresponding vectors or reduction lines (grey solid lines) are from Lewicka-Szczebak et al., (2016a), and represent minimum and maximum routes of isotopologue values with increasing $N_2O$ reduction to $N_2$ based on the reported range in the ratio between the isotope fractionation factors of $N_2O$ reduction for SP and $\delta^{18}O$ (Lewicka-Szczebak et al., 2016a).

**Tables**

**Table 1** Soil properties of the soil used in the experiment

**Table 2** The four saturation conditions used for the soil in the experiment



**Table 3** Contents of soil moisture, $NO_3^-$, $NH_4^+$ and C:N ratio and cumulative fluxes of $N_2O$ and $N_2$
and $CO_2$ from all treatments at the end of the incubation.
**Table 4** Scenarios with different combinations of $\delta^{18}O$ and SP endmember values and $\eta N_2O$-$N_2$
values to calculate maximum and minimum estimates of %$B_{DEN}$ (minimum, maximum and average
values adopted from Lewicka-Szczebak et al., (2016).
**Table 5** Ratios $N_2O$ / ($N_2O$ + $N_2$) for all treatments
**Table 6** The temporal trends in $\delta^{15}N_{bulk}$, $\delta^{18}O$, $\delta^{15}N_\alpha$, SP and %$B_{DEN}$ for all experimental treatments
**Table 7** Equations of fitted functions and correlation coefficients corresponding to Figure 5 for Site
Preference vs $\delta^{18}O$ in all treatments for three periods.



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



Table 1. Highfield soil properties

| Property | Units | Highfield | |
|---|---|---|---|
| Location | | Rothamsted Research Herts. | 930 931 932 933 |
| Grid reference | GB National Grid | TL129130 | 934 |
| | Longitude | 00°21'48"W | 935 |
| | Latitude | 51°48'18"N | 936 |
| Soil type | SSEW[a] group[c] | Paleo-argillic brown earth | 937 |
| | SSEW[a] series[d] | Batcombe | 938 |
| | FAO[bc] | Chromic Luvisol | 939 |
| Landuse | | Grass; unfertilised; cut | 940 |
| pH | | 5.63 | 941 |
| Sand (2000-63 µm) | g g$^{-1}$ dry soil | 0.179 | 942 |
| Silt (63-2 µm) | g g$^{-1}$ dry soil | 0.487 | 943 |
| Clay (<2 µm) | g g$^{-1}$ dry soil | 0.333 | 944 |
| Texture | SSEW[a] class[c] | Silty clay loam | 945 |
| Particle density | g cm$^{-3}$ | 2.436 | 946 |
| Organic matter | g g$^{-1}$ dry soil | 0.089 | 947 |
| Water content for packing | g g$^{-1}$ dry soil | 0.37 | |

[a]Soil Survey of England and Wales classification system
[b]United Nations Food and Agriculture Organisation World Reference Base for Soil Resources classification
system (approximation)
[c]Avery (1980)
[d]Clayden & Hollis (1984)



Table 2. The four saturation conditions set for the Highfield soil.

| Saturation condition | SAT/sat | HALFSAT/sat | UNSAT/sat | UNSAT/sat |
|---|---|---|---|---|
| Macropores | Saturated | Half-saturated | Unsaturated | Unsaturated |
| Micropores | Saturated | Saturated | Saturated | Half-saturated |
| *As prepared:* | | | | |
| Matric potential, -kPa | 4.1 | 12.3 | 27.3 | 136.9 |
| Water content, g 100 g$^{-1}$ | 47.7 | 42.5 | 37.2 | 29.4 |
| Water content, cm$^{-3}$ 100 cm$^{-3}$ | 61.1 | 54.4 | 47.7 | 37.3 |
| Water-filled pore space, % | 98 | 91 | 82 | 68 |
| Threshold pore size saturated, μm | 73 | 24 | 11 | 2 |
| *Final, following amendment:* | | | | |
| Matric potential, -kPa | 0 | 8.6 | 20.0 | 78.1 |
| Water content, g 100 g$^{-1}$ | 49.8 | 44.6 | 39.3 | 31.5 |
| Water content, cm$^{-3}$ 100 cm$^{-3}$ | 63.8 | 57.1 | 50.4 | 40.0 |
| Water-filled pore space, % | 100 | 94 | 85 | 71 |
| Threshold pore size saturated, μm | all | 35 | 15 | 4 |






Table 3. Contents of soil moisture, $NO_3^-$, $NH_4^+$ and C:N ratio and cumulative fluxes of $N_2O$ and $N_2$ and $CO_2$ from all treatments at the end of the incubation. Values in brackets are standard deviation of the mean of three values.

| Treatment | % Mean moisture | $NO_3^-$, mg N kg$^{-1}$ dry soil | $NH_4^+$, mg N kg$^{-1}$ dry soil | Total C, % | Total N, % | $N_2O$ kg N ha$^{-1}$ | $N_2$ kg N ha$^{-1}$ | Total emitted N | $CO_2$, kg C ha$^{-1}$ |
|---|---|---|---|---|---|---|---|---|---|
| SAT/sat | 39.8 (1.3) | 1.1 (0.4) | 104.3 (1.1) | 3.61 (0.04) | 0.35 (0.004) | 9.4 (1.1) | 54.0 (14.0) | 63.4 | 289.2 (30.4) |
| HALFSAT/sat | 40.2 (0.2) | 0.8 (1.0) | 104.2 (6.8) | 3.64 (0.08) | 0.36 (0.004) | 10.9 (0.4) | 51.7 (9.0) | 62.6 | 283.0 (35.5) |
| UNSAT/sat | 36.5 (2.1) | 51.2 (37.4) | 100.8 (5.7) | 3.64 (0.10) | 0.36 (0.007) | 23.7 (11.0) | 36.0 (28.5) | 59.7 | 417.6 (57.1) |
| UNSAT/halfsat | 34.3 (1.1) | 100.6 (16.1) | 71.3 (33.6) | 3.53 (0.08) | 0.36 (0.01) | 16.8 (15.8) | 17.2 (19.4) | 34.1 | 399.7 (40.6) |

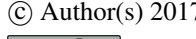

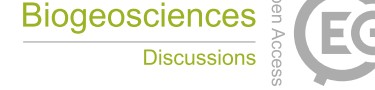

Table 4: Scenarios with different combinations of d$^{18}$O and Site Preference (SP) endmember values and η$_{N2O-}$
$_{N2}$ values to calculate maximum and minimum estimates of %Bden (minimum, maximum and average values
adopted from Lewicka-Szczabak et al., 2016a).

|  | SP0BD | SP0FDN | ηSP | η$^{18}$O |
|---|---|---|---|---|
| model (min endmember plus η) | -11 | 30 | -2 | -12 |
| model (max endmember plus η) | 0 | 37 | -8 | -12 |
| model (max endmember) | 0 | 37 | -5.4 | -12 |
| model (min endmember) | -11 | 30 | -5.4 | -12 |
| model (max η) | -5 | 33 | -8 | -12 |
| model (min η) | -5 | 33 | -2 | -12 |




Table 5. Ratios $N_2O / (N_2O + N_2)$ for all treatments

|  | SAT/sat | | HALFSAT/sat | | UNSAT/halfsat | | UNSAT/sat | |
|---|---|---|---|---|---|---|---|---|
| **Days** | mean | s.e. | mean | s.e. | mean | s.e. | mean | s.e. |
| -1 | 0.276 | 0.043 | 0.222 | 0.009 | 0.849 | 0.043 | 0.408 | 0.076 |
| 0 | 0.630 | 0.022 | 0.538 | 0.038 | 0.763 | 0.053 | 0.861 | 0.043 |
| 1 | 0.371 | 0.025 | 0.360 | 0.019 | 0.622 | 0.018 | 0.644 | 0.031 |
| 2 | 0.096 | 0.016 | 0.139 | 0.015 | 0.425 | 0.005 | 0.296 | 0.020 |
| 3 | 0.004 | 0.002 | 0.015 | 0.006 | 0.439 | 0.052 | 0.256 | 0.025 |
| 4 | 0.017 | 0.002 | 0.008 | 0.001 | 0.475 | 0.049 | 0.232 | 0.012 |
| 5 | 0.019 | 0.003 | 0.012 | 0.001 | 0.503 | 0.037 | 0.174 | 0.010 |
| 6 | 0.068 | 0.008 | 0.020 | 0.001 | 0.459 | 0.052 | 0.135 | 0.010 |
| 7 | 0.085 | 0.008 | 0.047 | 0.003 | 0.333 | 0.057 | 0.127 | 0.003 |
| 8 | 0.106 | 0.004 | 0.066 | 0.002 | 0.277 | 0.006 | 0.122 | 0.002 |
| 9 | 0.089 | 0.003 | 0.053 | 0.005 | 0.265 | 0.006 | 0.122 | 0.005 |
| 10 | 0.060 | 0.003 | 0.090 | 0.014 | 0.428 | 0.086 | 0.118 | 0.006 |
| 11 | 0.063 | 0.002 | 0.053 | 0.002 | 0.414 | 0.051 | 0.125 | 0.005 |





Table 6. The temporal trends in $\delta^{15}N_{bulk}$, $\delta^{18}O$, $\delta^{15}N_\alpha$, Site Preference (SP) and %$B_{DEN}$ for all experimental
treatments (values in brackets are the standard deviation of the mean)

| | $\delta^{15}N_{bulkAIR}$ (‰) | | | |
|---|---|---|---|---|
| **Day** | SAT/sat | HALFSAT/Sat | UNSAT/Sat | UNSAT/halfsat |
| -1 | -3.8 (2.1) | -6.2 (1.5) | -14.2 (10.9) | -23.6 (1.1) |
| 1 | -18.9 (1.6) | -25.5 (4.6) | -20.3 (2.6) | -20.8 (2.3) |
| 2 | -7.7 (4.2) | -12.7 (2.7) | -12.2 (2.0) | -13.9 (5.7) |
| 3 | -2.4 (1.8) | 14.0 (2.2) | -1.1 (7.6) | -4.4 (3.0) |
| 4 | -0.9 (2.2) | -0.3 (3.6) | -7.8 (4.6) | -9.3 (3.7) |
| 5 | -6.9 (0.9) | -4.3 (6.1) | -11.3 (3.7) | -8.9 (7.7) |
| 7 | -9.6 (1.5) | -10.0 (1.6) | -14.3 (4.7) | -13.4 (13.5) |
| 12 | -7.5 (1.2) | -8.6 (0.9) | -11.8 (2.6) | -21.3 (6.9) |
| | $\delta^{18}O_{SMOW}$ (‰) | | | |
| | SAT/sat | HALFSAT/Sat | UNSAT/Sat | UNSAT/halfsat |
| -1 | 33.3 (2.6) | 32.7 (3.0) | 31.4 (9.8) | 25.2 (4.9) |
| 1 | 42.9 (2.4) | 37.1 (3.8) | 32.3 (3.6) | 33.3 (2.1) |
| 2 | 54.0 (5.7) | 48.7 (4.5) | 42.7 (5.3) | 40.5 (5.0) |
| 3 | 45.7 (1.5) | 59.7 (3.2) | 53.4 (5.7) | 41.2 (1.0) |
| 4 | 42.5 (1.4) | 42.0 (3.7) | 38.1 (4.5) | 39.9 (7.7) |
| 5 | 36.0 (2.9) | 34.6 (3.7) | 30.4 (2.6) | 36.5 (6.9) |
| 7 | 32.2 (5.5) | 31.6 (5.5) | 28.4 (4.4) | 32.7 (5.4) |
| 12 | 34.9 (5.6) | 34.1 (2.7) | 32.4 (2.9) | 28.5 (5.0) |
| | $\delta^{15}N\alpha_{AIR}$ (‰) | | | |
| | SAT/sat | HALFSAT/Sat | UNSAT/Sat | UNSAT/halfsat |
| -1 | -0.3 (3.4) | -2.6 (1.8) | -9.5 (12.0) | -19.7 (2.1) |
| 1 | -17.4 (1.8) | -24.0 (5.8) | -20.2 (2.0) | -21.1 (2.6) |
| 2 | -4.6 (4.2) | -9.5 (3.6) | -11.1 (1.1) | -13.8 (5.9) |
| 3 | -0.8 (1.3) | 17.2 (4.0) | 7.6 (4.7) | -2.7 (3.2) |
| 4 | 1.0 (2.5) | 0.7 (2.2) | -3.5 (3.7) | -2.8 (7.7) |
| 5 | -5.9 (0.7) | -2.9 (5.4) | -9.4 (3.9) | -5.2 (7.9) |
| 7 | -7.8 (2.3) | -5.3 (4.2) | -12.3 (5.6) | -7.7 (11.5) |
| 12 | -3.3 (2.1) | -4.6 (0.6) | -8.1 (4.2) | -15.3 (5.5) |
| | $SP_{AIR}$ | | | |
| | SAT/sat | HALFSAT/Sat | UNSAT/Sat | UNSAT/halfsat |
| -1 | 7.0 (3.9) | 7.1 (4.2) | 9.4 (2.1) | 7.7 (1.9) |
| 1 | 2.9 (0.6) | 3.0 (2.3) | 0.1 (1.8) | -0.7 (1.4) |
| 2 | 6.3 (0.64) | 6.4 (1.9) | 2.2 (2.0) | 0.2 (1.9) |
| 3 | 3.3 (1.0) | 6.4 (6.9) | 11.9 (12.4) | 5.9 (0.8) |
| 4 | 3.7 (0.6) | 2.0 (6.2) | 8.7 (5.9) | 5.4 (3.0) |
| 5 | 2.0 (0.4) | 3.0 (2.1) | 3.9 (0.5) | 7.4 (2.3) |
| 7 | 5.0 (2.1) | 9.2 (5.2) | 3.9 (1.8) | 11.2 (4.1) |
| 12 | 8.4 (3.3) | 7.9 (0.8) | 7.3 (3.7) | 11.8 (5.3) |
| | Estimated range of %$B_{DEN}$ | | | |
| | SAT/sat | HALFSAT/sat | UNSAT/sat | UNSAT/halfsat |
| -1 | 63-100 | 60-100 | 53-85 | 56-84 |
| 1-2 | 68-100 | 67-100 | 73-100 | 77-100 |
| 3-12 | 78-100 | 79-100 | 60-100 | 54-86 |



Table 7. Equations of fitted functions and correlation coefficients corresponding to Figure 5 for Site
Preference (SP) vs $\delta^{18}$O in all treatments for three periods.

| Treatment | Days 1-2 | Days 3-5 | Days 7-12 |
|---|---|---|---|
| SAT/sat | y = 0.2151x - 5.8386, R² = 0.6529 | y = 0.1204x - 1.848, R² = 0.397 | y = 0.5872x - 12.223, R² = 0.985 |
| HALFSAT/sat | y = 0.3447x - 10.129, R² = 0.9048 | y = 0.23x - 7.0689, R² = 0.2188 | y = 0.4063x - 6.2632, R² = 0.6876 |
| UNSAT/sat | y = 0.2709x - 8.9968, R² = 0.8664 | y = 0.7248x - 18.874, R² = 0.507 | y = 0.6848x - 15.236, R² = 0.7156 |
| UNSAT/halfsat | y = -0.0146x + 0.2506, R² = 0.0024 | y = 0.3589x - 7.2194, R² = 0.4839 | y = -0.318x + 21.261, R² = 0.1491 |










1b.




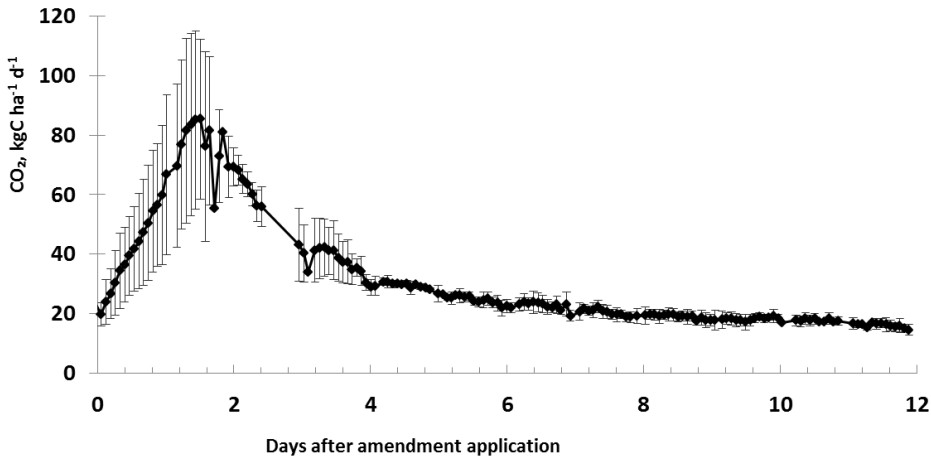



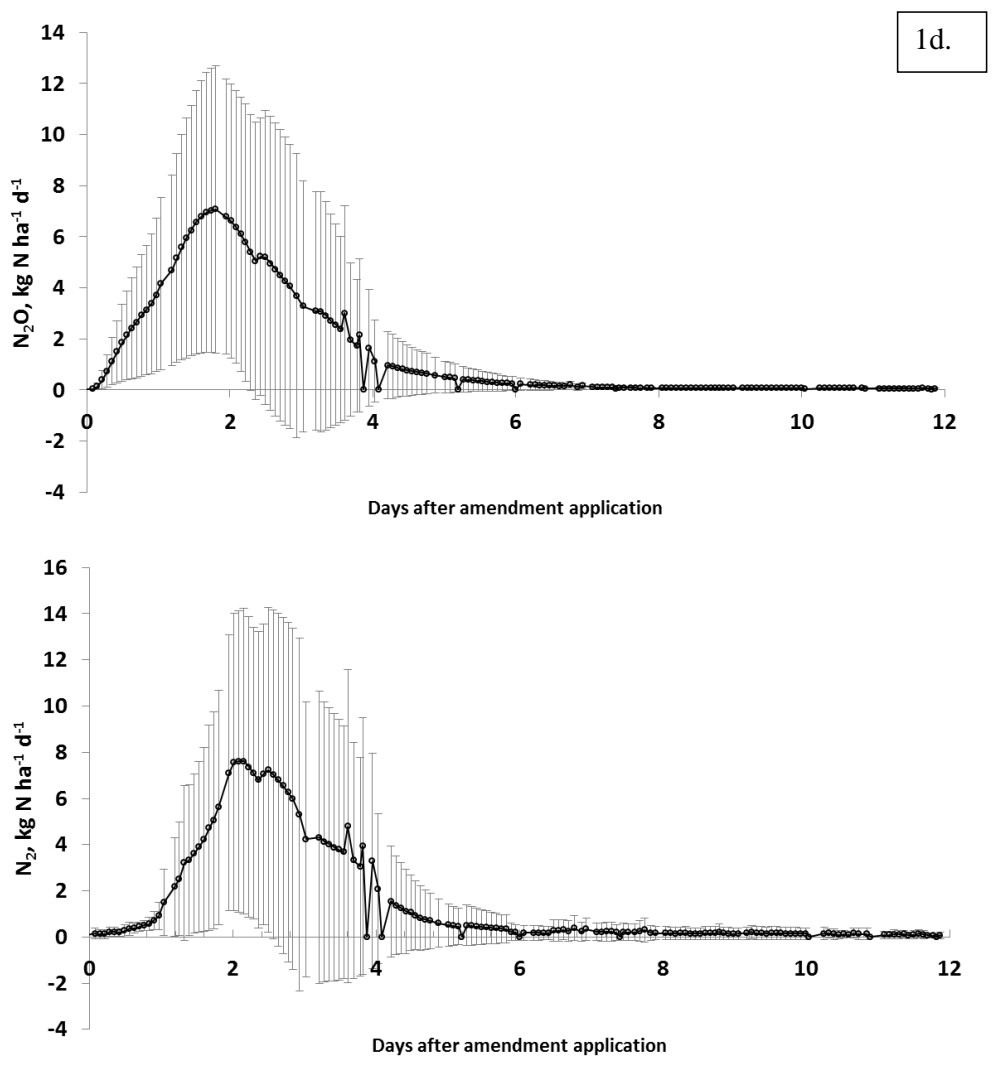

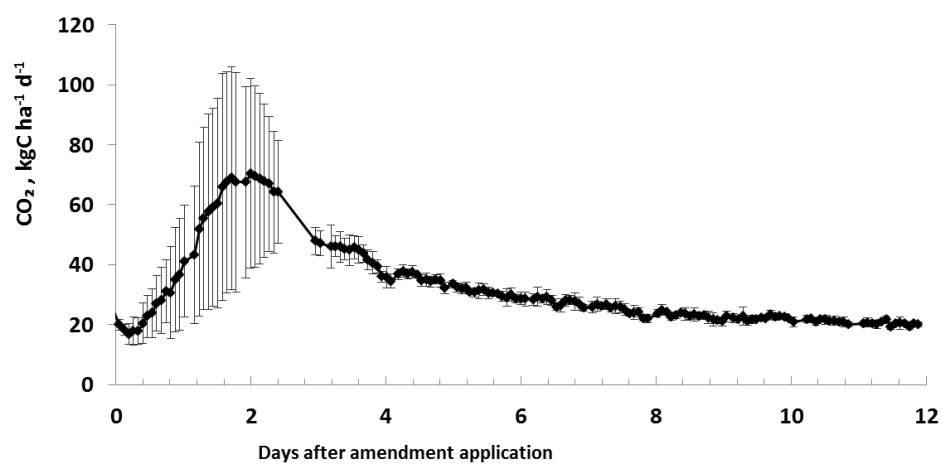



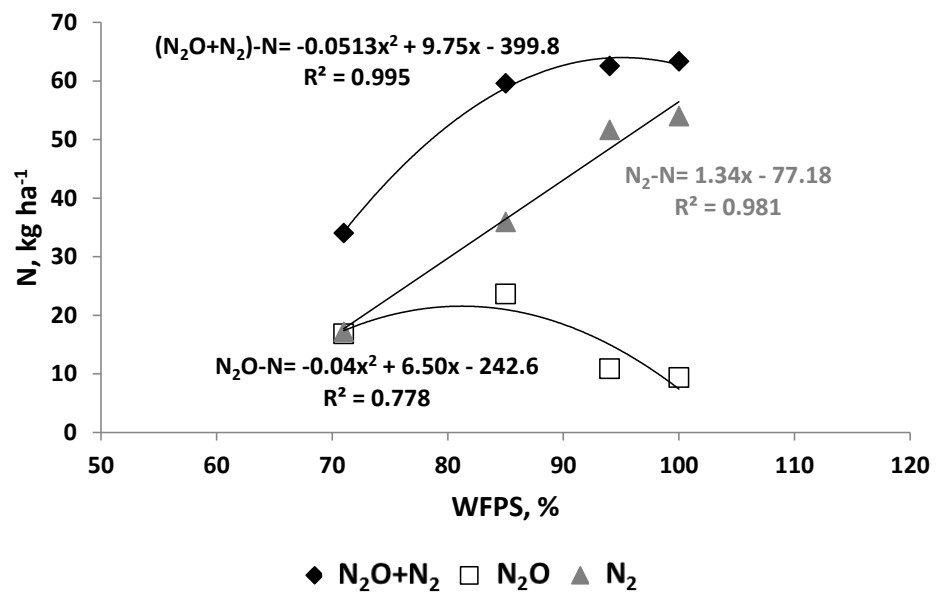

Figure 2





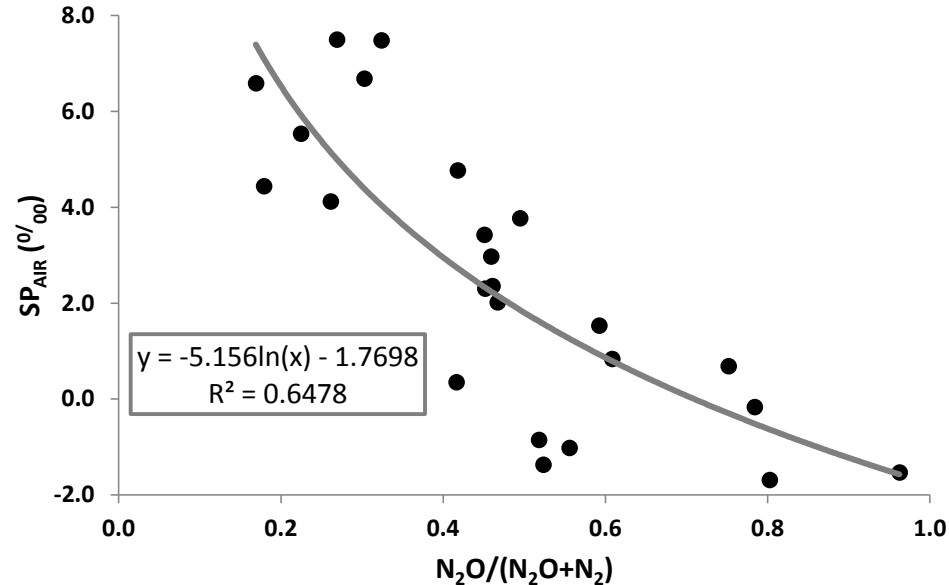

Figure 3





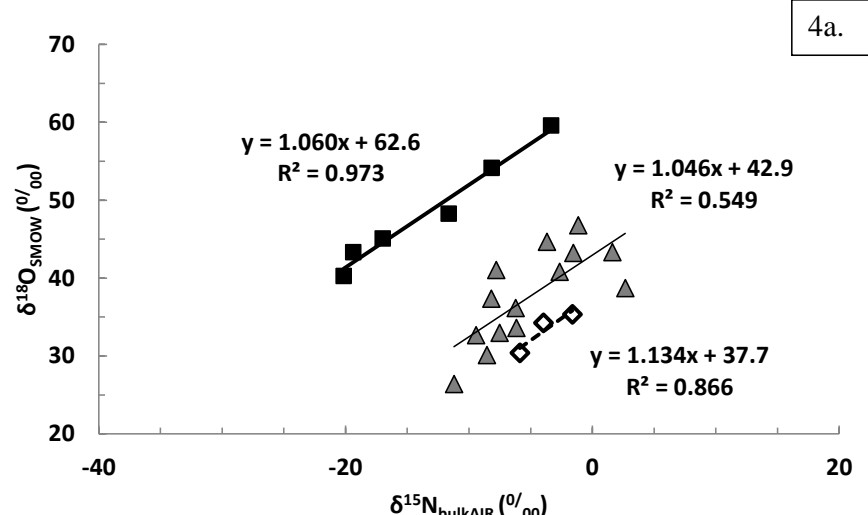

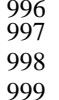


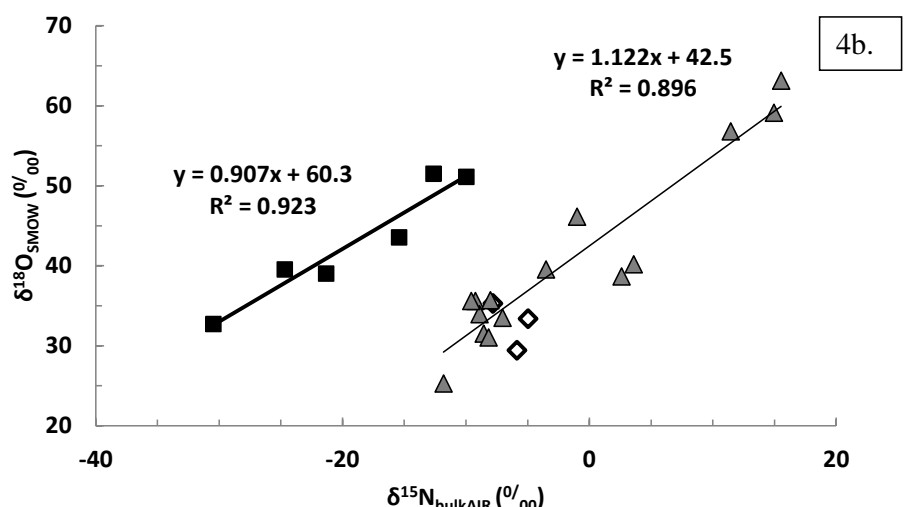






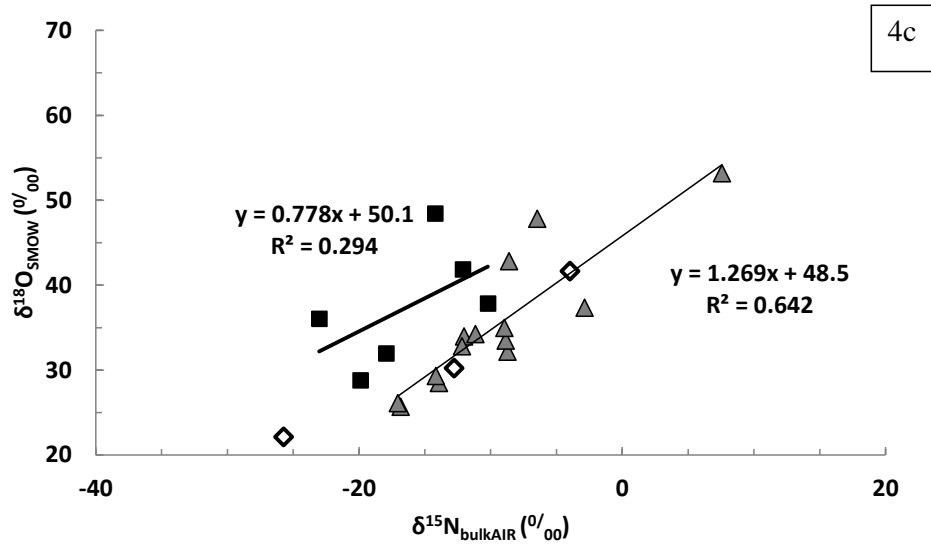


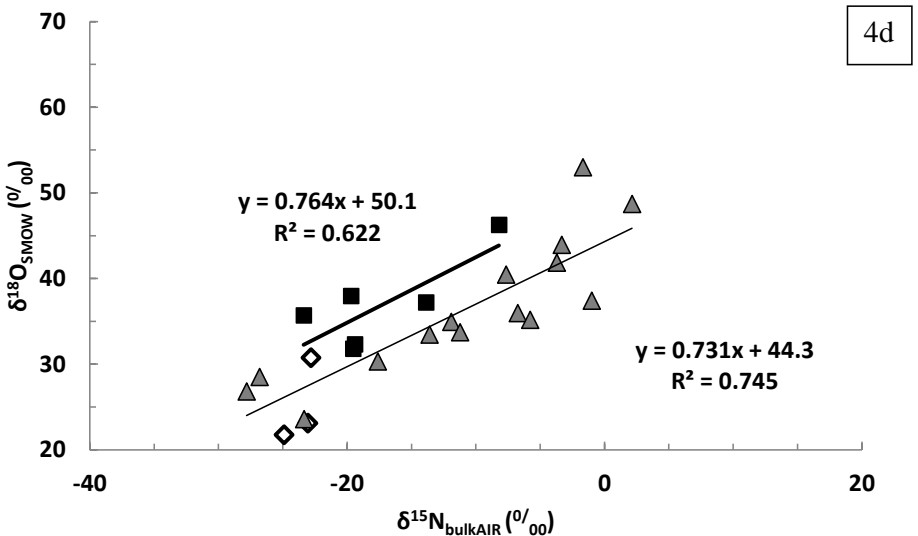




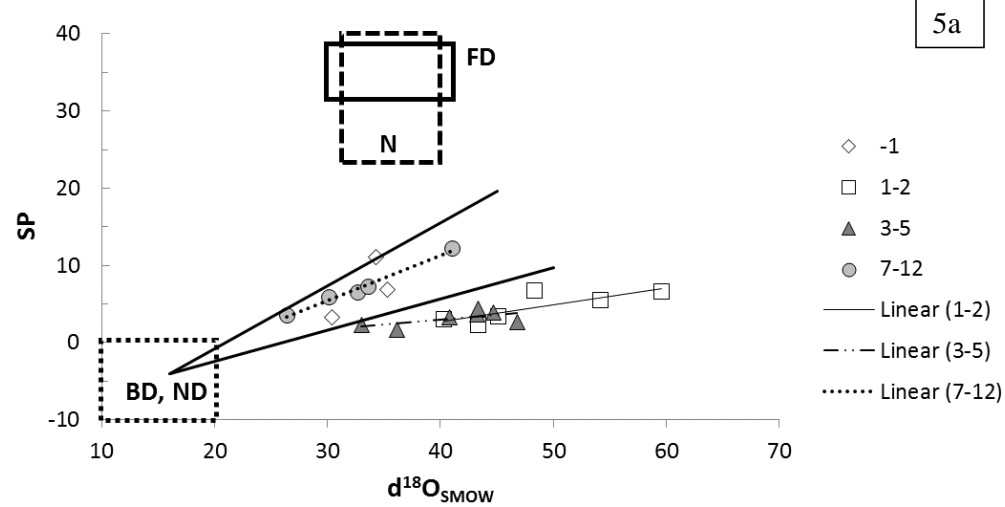


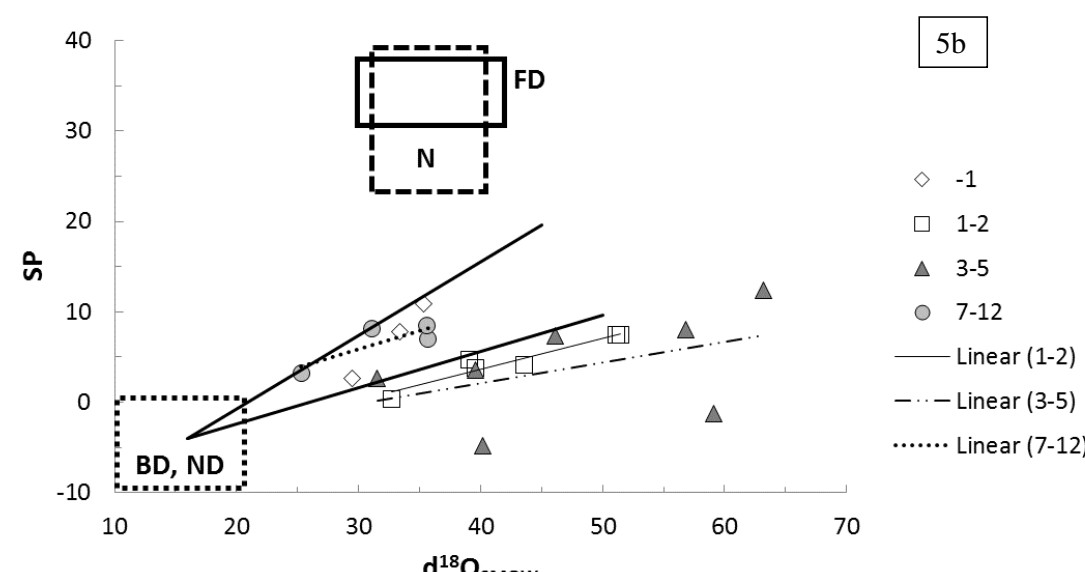






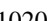


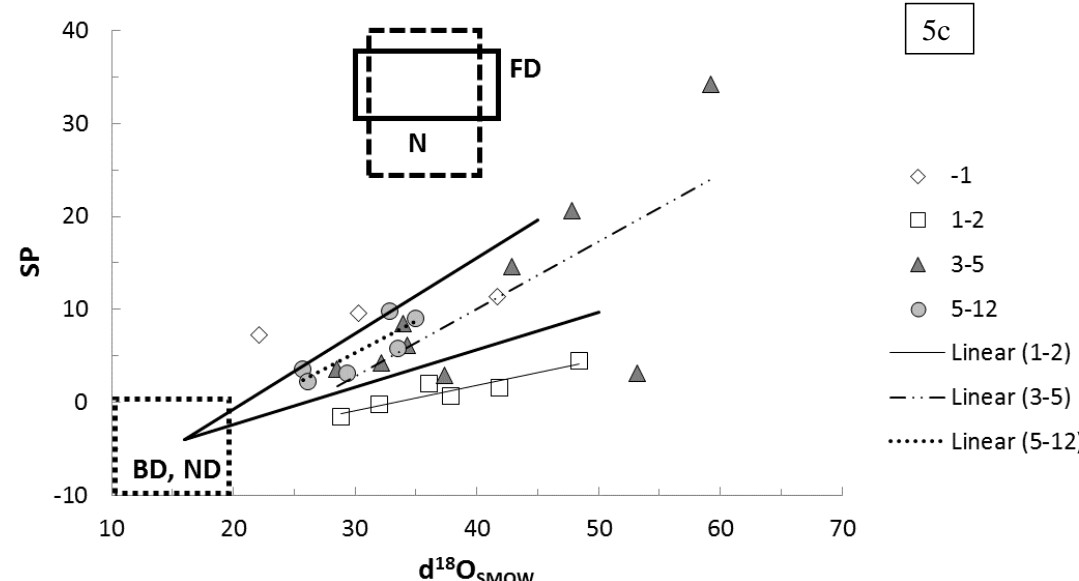


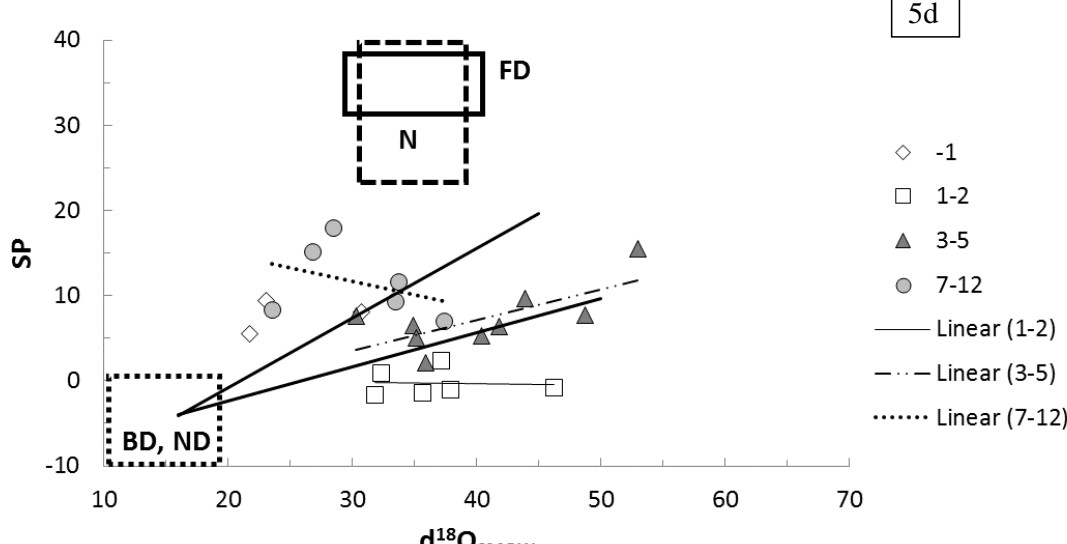
