# Peer review of "Effect of soil saturation on denitrification in a grassland soil"

_Biogeosciences, 2016_

## Referee Comment (RC1) · Anonymous Referee #3 · 10 Feb 2017

This is an interesting study that addresses the roles of soil compaction and water saturation levels on N2O production and the microbial origins of N2O. The results are not terribly profound but this is an important contribution to the literature as the precise causes of N2O hot spot production are still unresolved. Overall I found the writing to suffer from incorrect grammar and English writing style. Further, the manuscript is much longer than it needs to be. The manuscript would greatly benefit from a major rewrite and could be re-written as a short concise note rather than a full research paper. I've identified some issues with the writing below but there are numerous problems beyond what I have listed.

Line 26 to 29: As this sentence contains both a colon and a semi-colon it needs to be broken into at least two sentences. I do not understand the meaning of the portion after the colon (28-29).

Line 73 and 74: Please check with Coplen (2011) regarding the correct usage of "iso-topologues" and " isotopomers".

Line 97-98: Why is "soil volume" the key control on the net isotope effect? This seems more like an experimental condition rather than a governing soil process.

Line 111-112: Generally avoid one-sentence paragraphs. This statement belongs more appropriately in the Methods section and could be deleted here.

Line 159: This paragraph is much longer and more detailed than it needs to be.

Line 323-324: Use past tense here.

Line 338: Delete "already".

Line 351: Incorrect word use. SP values don't "show"; rather they are obtained. Use past tense to describe trends in the experimental data throughout this paragraph.

Line 363: Don't describe "the plot"; rather simply refer to the trends between the parameters.

Line 365: Regressions don't suggest but simply describe a (presumably significant) relationship between two parameters. You can state that the intercept of the regression equation relating SP and the N2O/(N2O+N2) was – 2 per mil.

Line 367-369: The writing is confusing here; I can not follow the meaning of this sentence.

Line 370: It is not helpful to refer to data in a figure of another paper. Describe the main significance to the similarity between these data sets. Line 374: Again, don't state what is plotted in Figure 4, describe the relationships between the variables and refer to the figure.

Line 383: The r2 values by themselves are not very relevant. What is relevant is if the relationships are significant and their associated p values.

Line 389: See comment for line 374.

Tables 1, 4, 5 and 6: These tables could readily be placed in the Supplementary Documents.

Figure 5: These figures are not well organized. Put a box around the legends so that we know they are legends. Within the legend, the line should be placed through the data points rather than defining each line as "Linear". The y-axis title should display delta not "d".

---

## Referee Comment (RC2) · Anonymous Referee #1 · 21 Feb 2017

General comments:

The paper aims to quantify $N_2O$ and $N_2$ production process in grassland soils and its dependence on compaction. $N_2O$ and $N_2$ emissions and their isotopic signature have been monitored over a period of 12 days after amendment of $KNO_3$. The presented laboratory studies simplify the complex soil pore system into macro and micropores and uses four stages in a rather narrow range of 70 to 95% "mean" WFPS.
The experimental setup is described in detail. The results agree with the expected values, i.e. domination of bacterial denitrification processes for the higher water content and an increasing share of other contribution for when part of the pores is dry. The measurement of the isotopic signature allows to distinguish different production processes and their dependence on the water status of the macro and micropores.

I had difficulties to follow the argumentation and get quickly lost in too many in details. I also miss a discussion of the significance of the presented findings for the characterization of the emissions of N-species for real grassland systems, although in the introduction (e.g. lines 62 and 63) the study is set in this context.

The used soil stem from a long-term permanent grassland. But the preparation of the samples (a necessary step for the laboratory study) destroys the specific characterization of a grassland soil. Roots and the organization of the aggregates are removed and there is no plant growth that greatly influence the distribution and availability of N substrate as well as the oxygen supply. It should also be mentioned that a large share of N-input in agricultural system occurs in reduced N-form (excrement's, urea or ammonium nitrate). In grazed system, spatial heterogeneity is related to the urine patches with a very high N-input on a very limited area. Also, compaction (trampling by animal, tractor tracks) is spatially very heterogeneous and likely uncoupled to N-substrate input.

The results from the present study shows for $N_2O$ as well as ($N_2O$ and $N_2$) emission a remarkably low variability among the four treatment, much lower as typically experienced in field measurements.

Below are given specific comments as a guideline to improve the manuscript

Abstract:

*Lines 16 and 17: The soil emitted $N_2O$ is predominantly derived from denitrification and to a smaller extent, nitrification in soils,*

This is a too crude generalization. There are many ways to produce $N_2O$ and the share between them depends in a complex manner from the main driver, such as oxygen content, substrate availability, etc.

*Lines 20 and 21: Soil water content expressed as water filled pore space (WFPS) is a major controlling factor of emissions and its interaction with compaction, has not been studied at the micropore scale.*

This is slightly misleading as the experimental setup can only measure net fluxes across the surface of the entire soil samples and naturally does not allow to determine $N_2O$ production/consumption in and out of the micropores.

Introduction

*Lines 210 and 211: concentration) for 24 h, or until the system and the soils atmosphere were emitting low background levels of both $N_2$ and $N_2O$ ($N_2$ can get down to levels of 280 ppm much smaller than atmospheric values).*

Please indicate these „background" values.

*Lines 222 and 223: Flushing was carried out with He for half an hour before the solution was required for application to the soil cores and continued during the application process to avoid atmospheric N2 contamination (a total of one and a half hours).*

How this affects the oxygen availability?

*Lines 304 and 305: We accepted these as unavoidable features of the experimental set-up, but we suggest that the main response of the gaseous emissions occurred under the initial conditions, prior to the loss of water over subsequent days.*

"We suggest" is a strange formulation, either the time coarse of the emissions clearly shows this, or it is an assumption.

Results

*Lines 311 UNSAT/halfsat (50-100 N kg- dry soil)*

Unit of $NO_3^-$ seems incorrect. Also, the header of Table 2 is wrong (twice UNSAT/SAT)

*Lines 349 to 351: The results showed that the total N emission ($N_2O+N_2$) (Table 3) had a consistent decreasing trend, with decreasing soil moisture i.e. from 63.4 for SAT/sat (100% WFPS) to 34.1 kg N ha-1 (71% 351 WFPS) for UNSAT/halfsat.*

I don't see a consistent decreasing trend. Only the driest treatment shows a lower emission.

It also would make more sense to use the same reference for the mineral N content as well as the cumulative gaseous emissions (e.g. per g soil).

*Lines 351 and 352: The maximum cumulative $N_2O$ occurred at around 80% WFPS as Fig. 2 shows.*

This is an overinterpretation. There are four values and a fit with three unknown is applied.

Noticeable emissions of $N_2O$ and $N_2$ occur in all four treatment only up to day four. Bacterial denitrification is identified as the main production pathway. This is due to the experimental setup with a combined amendment of $KNO_3$ and glucose, a setup that produce good conditions for denitrification irrespective of the specific treatment.

---

## Referee Comment (RC3) · Anonymous Referee #2 · 22 Feb 2017

General remarks

This paper presents results from a sophisticated laboratory experiment in which an agricultural soil was compacted and adjusted to 4 different moisture conditions. Glucose and nitrate was added and the formation, isotopic and isoptomeric composition of gasesous N was measured over a period of 12 days. Using those data the authors try to determine the contribution of different processes to N gas formation. The paper is a good example how much information you can get from experimental data if you spend a lot of energy in calculations and data analysis. However, in my eyes the paper has three critical weaknesses:

1.) The results are not really new. It is known for a long time that addition of nitrate and glucose stimulates denitrification in soils and that denitrification is favored under wetter

conditions. All the points in the conclusions are not new. If there is new knowledge obtained from the study, it has to be elaborated more clearly.

2.) The paper is lacking a clear story. It is not really clear to me what was the final purpose of all those detailed analysis. There are some hypothesis mentioned at the end of the introduction but the rest of the manuscript is not tailored to address those hypotheses. The hypothesis that wetter conditions reduce heterogeneity could be answered from just looking at the error bars in figure 1 – you do not need sophisticated analysis to prove this point. Aiming to understand what is going on in one′s own experiment (as stated in the last sentence of the introduction) is not a sufficient aim of a paper.

3.) There are some problems with the experimental approach which limit interpretation of the data. First, moisture conditions were not constant but changed a lot during the experiment. The second treatment, for example at the end of the experiment had the same water content as the third treatment in the beginning. They had changing substrate concentrations in parallel to changing moisture conditions. Thus, the interpretation of moisture effects during the course of the experiment is difficult. A way to minimize that effect would have been to moisten the supplied He/O2 gas. I would also expect that water loss was highest in the beginning, when the surface layer was drying. A way to get some information about temporal changes of water content would have been to weigh the incubation vessels during the incubation. Second, they measured gas emission – not gas production. They mention this problem in the paper but somehow ignore its consequences. The emitted gas probably originates from those sites which are physically linked to the atmosphere, while gas production, e.g. in the center of aggregates did probably contribute less to the emitted gas. So, the conclusions drawn from the analysis could be valid only for a part of the soil volume.

Detailed comments

l.17: remove "soils" l.40: What do you mean with "benign" for the environment. Do you mean the process is important because it closes the global N cycle because it reverses

N-fixation? l.64-73: I would move this paragraph to an earlier point, before talking about compaction. l.72: I would replace "powerful tool" by "basis". l.81: If there are several references for one statement, present them in chronological order. l.81-82: Remove sentence l.83: "…under the conditions…" l.92: Be more specific. What do you mean by "other steps of denitrification"? l.93: "reported here". l.100: Does that mean that those results are only relevant at elevated C and N? l.108: Why CO2? l.112: "controlled laboratory conditions" l.119: What do you mean by "heterogeneity in N emissions"? l.120: I am not a soil scientist, but is that really new? l.121: Aiming to understand what is going on in one's own experiment is not a sufficient aim of a paper. l.137: Verb missing. "was applied"? l.228: "CO2 was measured…" l.230: replace "pulled together in one sample" by "pooled" l.232: Remove sentence. There is a similar sentence in the results section. l.268: Were the data normal distributed? l.275: "mixing model was then used" (use past tense) l.283: When did this occur and what is a possible explanation? Wrong fractionation factors? l.290: A TCD is an detector – not an analyzer. l.303 Why was the gas stream not bubbled through water to saturate it with water? l.305: I would expect the highest water loss right in the beginning. l.306. But they were similar between treatments in the end although different starting conditions. l.314-316: There was a high variability in the data. l.318: Remove "The results showed that" l.329: I do not see that in Figure 1. In Unsat/sat the N2O maximum was at 12 kg N/ha d, not around 7. l.348. Right. But what are the consequences of this for your experiment and its interpretation? l.354: You probably mean "CO2 fluxes". Why was CO2 measured? l.360: The carbon budget is interesting but complicated. Could you calculate recovery rates for the added glucose? It looks as if there are recoveries higher than 100%. Can this be interpreted as a priming effect? A problem with using CO2 for carbon budgeting is, that depending on pH you also have other IC species in the soil solution. Do you know the pH in your soils? l.370: Add article before "period" l.375: The SP data have a high standard deviation. Are the differencers discussed in this paragraph real? l.391: You may consider adding these data to the plot. l.394: Separate into two sentences. Start second one with "In our data, maximum …." l.404 So what is the message of

this paragraph with respect to the first sentence of the paragraph? l.405: Why was this done? l.428: Why was this plot done? l.441: I do not see data within those areas in the plots. l.456: "sat" page 19: It is difficult to detect the storyline on this page. L513: Could it be that there was C limitation in the dryer treatments because glucose was metabolized aerobically? l.534-537: The message of the $CO_2$ paragraph is not really clear. Are the $CO_2$ data helpful in this manuscript? l.539: How much is the unacounted N-loss in comparison to the accounted gasesous losses? l.541: NO: What are typical NO fluxes in the literature? Can the NO flux have a significant magnitude? The same applies to microbial biomass: Is the microbial biomass potentially formed from the unaccounted N-loss in a realistic order of magnitude? l.567: How should nitrification contribute to BDEN? Do you mean nitrifier-denitrification? l.636: I do not understand the content and purpose of this paragraph. l.719: Don't you have 4 periods in the figure? Table 3: Unit missing for Total emitted N. Tables 5 and 6: I wonder whether these data could be presented better in figures. Figure 5: the four sub-graphs are quite similar. Isn't a conclusion that the results were not much influenced by soil moisture? Do you really need 4 graphs?

---

## Author Comment (AC1) · 27 Apr 2017

This is an interesting study that addresses the roles of soil compaction and water saturation levels on N2O production and the microbial origins of N2O. The results are not terribly profound but this is an important contribution to the literature as the precise causes of N2O hot spot production are still unresolved. Overall I found the writing to suffer from incorrect grammar and English writing style. Further, the manuscript is much longer than it needs to be. The manuscript would greatly benefit from a major rewrite and could be re-written as a short concise note rather than a full research paper. I've identified some issues with the writing below but there are numerous problems beyond what I have listed. R: the majority of the authors consist of native English speakers and the English has been revised by them, so we believe the quality of the English is good. We think that providing the current level of detail in this manuscript as

a full research paper is required to give further evidence for the need to use isotopic signatures and modelling approaches of N2O in order to describe the driving source processes of this gas as emitted from soils. Line 26 to 29: As this sentence contains both a colon and a semi-colon it needs to be broken into at least two sentences. I do not understand the meaning of the portion after the colon (28-29). R: thanks for the suggestion, paragraph has been split.

Line 73 and 74: Please check with Coplen (2011) regarding the correct usage of "iso-topologues"and " isotopomers".

R: we have now modified the text according to Coplen's definitions below and used iso-topocule always if SP AND d18O are addressed, isotopomer if ONLY SP is addressed. According to Coplen: 'The molecular species can be an isotopologue, an isotopomer, or neither. For example, the three molecular species 15N2 16O, 14N15N16O, and 15N14N16O are isotopocules, but they are neither isotopologues (because the latter two do not differ in isotopic composition) nor isotopomers (only the latter two are isotopomers). Isotopolog: Molecular species that differ only in isotopic composition (number of isotopic substitutions) and relative molecular Mass. Isotopomers: Molecular species having the same number of each isotopic atom (thus, the same relative molecular mass) but differing in their positions.' We defined these in the introduction as: 'Isotopologues of N2O represent the isotopic substitution of the O and/or the two N atoms within the N2O molecule. The isotopomers of N2O, are those differing in the peripheral ($\beta$) and central N-positions ($\alpha$) of the linear molecule' which we believe agree with the definition given by Coplen.

Line 97-98: Why is "soil volume" the key control on the net isotope effect? This seems more like an experimental condition rather than a governing soil process. R: we changed the text for: "The results generally confirmed the range of values of $\eta$ (net isotope effects) and $\eta$18O/$\eta$15N ratios reported by previous studies for N2O reduction for that part of the soil volume were denitrification was enhanced by the N+C amendment. This did not apply for the other part of the soil volume not reached by the N+C

amendment."

Line 111-112: Generally avoid one-sentence paragraphs. This statement belongs more appropriately in the Methods section and could be deleted here. R: text has been moved as suggested

Line 159: This paragraph is much longer and more detailed than it needs to be. R: section has been moved to a supplementary material.

Line 323-324: Use past tense here. R: all throughout this section (3.2) there is only past tense. I am not sure where the reviewer refers to.

Line 338: Delete "already". R: deleted as suggested.

Line 351: Incorrect word use. SP values don't "show"; rather they are obtained. Use past tense to describe trends in the experimental data throughout this paragraph. R: text has been amended.

Line 363: Don't describe "the plot"; rather simply refer to the trends between the parameters. R: text amended.

Line 365: Regressions don't suggest but simply describe a (presumably significant) relationship between two parameters. You can state that the intercept of the regression equation relating SP and the N2O/(N2O+N2) was – 2 per mil. R: changes have been introduced.

Line 367-369: The writing is confusing here; I cannot follow the meaning of this sentence. R: These are the lines in the submitted pdf: "This is in juxtaposition with the situation when the N emissions are dominated by N2 or N2O is low, where the SP values of soil emitted N2O were much higher (Fig. 3), pointing to an overall product ratio related to an 'isotopic shift' of 10 to 12.5o/oo." We modified to (including previous sentence): "The plot of the N2O / (N2O + N2) ratio vs SP for all treatments in the first two days (when N2O was increasing and the N2O / (N2O + N2) ratio decreasing) shows a significant negative response of the SP when the ratio increased (Fig. 3).

The regression suggests that when the emitted gaseous N is dominated by N2O (ratio close to 1) the SP values will be slightly negative with values around -2 (Fig. 3), i.e. within the range SP range of bacterial denitrification. With decreasing N2O / (N2O + N2) ratio the SP values of soil emitted N2O were increasing to values up to 8 per mil."

Line 370: It is not helpful to refer to data in a figure of another paper. Describe the main significance to the similarity between these data sets. R: I think the reviewer here refers to line 389. We are not referring to a figure necessarily but to the data from Lewicka-Szczebak et al. (2014). The significance was explained in the discussion: 'These results confirm from 2 independent studies Lewicka-Szczebak et al., 2014) that there is a relationship between the product ratios and isotopic signatures of the N2O emitted.'

Line 374: Again, don't state what is plotted in Figure 4, describe the relationships between the variables and refer to the figure. R: This is in line 406. We have edited the text as suggested.

Line 383: The r2 values by themselves are not very relevant. What is relevant is if the relationships are significant and their associated p values. R: R2 are reported in lines 412 onwards. We have analysed the regressions and introduced the P values as suggested.

Line 389: See comment for line 374. R: I think reviewer refers to line 428. We have stated the new figure was done similarly to the previous one, so we have left the text as it was.

Tables 1, 4, 5 and 6: These tables could readily be placed in the Supplementary Documents. R: yes, it would be possible, but we would like to have the editor's view before moving them.

Figure 5: These figures are not well organized. Put a box around the legends so that we know they are legends. Within the legend, the line should be placed through the

data points rather than defining each line as "Linear". The y-axis title should display delta not "d". R: Legends have now been enclosed by a box. The 'Linear' word in the legend clarifies that a linear function was fitted so we have left this as it was. The reviewer refers to the X axis, delta has been changed.

Please also note the supplement to this comment:
http://www.biogeosciences-discuss.net/bg-2016-556/bg-2016-556-AC1-supplement.pdf

**Supplement:**

Supplement 1

The gravimetric soil water release characteristic for the soil, as given in Gregory *et al.* (2010) represents the assumed pore size distribution, and a fitted van Genuchten function (van Genuchten, 1980) with the Mualem (1976) constraint of m = 1–1/n):

$$\theta_h = \theta_r + \frac{\theta_s - \theta_r}{[1 + (\alpha h)^n]^{1 - \frac{1}{n}}} \qquad [1]$$

where $\theta_s$, $\theta_r$ and $\theta_h$ are the saturated, residual (water content at permanent wilting point) and *h* matric potential gravimetric water contents (g g$^{-1}$), respectively; *h* is the matric potential (|kPa|, i.e. the absolute value), $\alpha$ is a fitted parameter approximating the inverse of *h* at the inflection point (|kPa|$^{-1}$), often linked to the air-entry point, and m and n are dimensionless fitted parameters related to the shape of the function.

The somewhat arbitrary saturation state known as "field capacity" represents the idealised condition UNSAT/sat, where the macropores have drained and the micropores have yet to drain. As field capacity has typically corresponded to a matric potential anywhere between -5 to -33 kPa, we chose -20 kPa as our UNSAT/sat condition, where the threshold pores size between water-filled pores at this matric potential is 15 µm. The matric potential corresponding to SAT/sat was obviously 0 kPa, to give full saturation of all the pores. To calculate the intermediate HALFSAT/sat condition, we took the mid-point gravimetric water content between 0 and -20 kPa from the water release characteristic, and calculated the corresponding matric potential using Eq. [1], which was -8.6 kPa. We also calculated the mid-point gravimetric water content between that at -20 kPa and $\theta_r$, and found the corresponding matric potential (Eq. [1]) to be -78.1 kPa. We used this to represent the UNSAT/halfsat condition. As $\theta_r$ was non-zero (in fact it was 0.236 g g$^{-1}$), due to the fine-textured nature of the soil, we accept that at -78.1 kPa the micropores were not truly half-saturated but would have been in a wetter condition than this. However due to our method for equilibrating the

soils prior to experimentation, we required a suitable matric potential not lower than -1500 kPa that we could control in the laboratory (see below). It could be argued that trying to attain a water content in the hygroscopic range (that held at potentials much lower than -1500 kPa, often in the vapour phase), where the true mid-point water content between that at -20 kPa and complete dryness in this soil lay, was not especially relevant to denitrification processes expected in such a soil. There was one final adjustment to make. The subsequent incubation experiment was to involve a 15 ml (3 × 5 ml) addition of solution (see below). Through knowing masses and volumes of the solid-water-air phases of our blocks, we therefore calculated revised matric potentials which would mean that the subsequent addition of solution would achieve the target potentials given above. The target matric potentials of 0 (SAT/sat), -8.6 (HALFSAT/sat), -20.0 (UNSAT/sat) and -78.1 kPa (UNSAT/halfsat) were revised to -4.1, -12.3, -27.3 and -136.9 kPa, respectively (see summary in Table 2).

---

## Author Comment (AC2) · 27 Apr 2017

The paper aims to quantify N2O and N2 production process in grassland soils and its dependence on compaction. N2O and N2 emissions and their isotopic signature have been monitored over a period of 12 days after amendment of KNO3. The presented laboratory studies simplify the complex soil pore system into macro and micropores and uses four stages in a rather narrow range of 70 to 95% "mean" WFPS. The experimental setup is described in detail. The results agree with the expected values, i.e. domination of bacterial denitrification processes for the higher water content and an increasing share of other contribution for when part of the pores is dry. The measurement of the isotopic signature allows to distinguish different production processes and their dependence on the water status of the macro and micropores. I had difficulties to follow the argumentation and get quickly lost in too many in details. I also

miss a discussion of the significance of the presented findings for the characterization of the emissions of N-species for real grassland systems, although in the introduction (e.g. lines 62 and 63) the study is set in this context. The used soil stem from a long-term permanent grassland. But the preparation of the samples (a necessary step for the laboratory study) destroys the specific characterization of a grassland soil. Roots and the organization of the aggregates are removed and there is no plant growth that greatly influence the distribution and availability of N substrate as well as the oxygen supply. It should also be mentioned that a large share of N-input in agricultural system occurs in reduced N-form (excrement's, urea or ammonium nitrate). In grazed system, spatial heterogeneity is related to the urine patches with a very high N-input on a very limited area. Also, compaction (trampling by animal, tractor tracks) is spatially very heterogeneous and likely uncoupled to N-substrate input. R: the authors agree that soil structure is destroyed, but as the referee says himself, this is a laboratory study, so we are not trying to reproduce the field conditions but to understand soil processes. In fact, we are assessing the potential for this soil to emit N2O and for this reason we have optimised the conditions for denitrification. The plant is not included for the same reason, as we aim to understand the processes in the soil, although we agree that the plant plays a major role in modifying these processes. The soil used in this study is not sourced from a grazed grassland, but a grassland that is cut, so the effect of the animal, via grazing, soil compaction and excreta deposition is not relevant. The results from the present study shows for N2O as well as (N2O and N2) emission a remark-ably low variability among the four treatment, much lower as typically experienced in field measurements. Below are given specific comments as a guideline to improve the manuscript Abstract: Lines 16 and 17: The soil emitted N2O is predominantly derived from denitrification and to a smaller extent, nitrification in soils, This is a too crude generalization. There are many ways to produce N2O and the share between them depends in a complex manner from the main driver, such as oxygen content, substrate availability, etc. R: the authors agree with the referee point and in fact the sentence goes on to say: 'both processes controlled by environmental factors and their interac-

tions, and are influenced by agricultural management'. We have however made it clear that it is a generalisation. Lines 20 and 21: Soil water content expressed as water filled pore space (WFPS) is a major controlling factor of emissions and its interaction with compaction, has not been studied at the micropore scale. This is slightly misleading as the experimental setup can only measure net fluxes across the surface of the entire soil samples and naturally does not allow to determine N2O production/consumption in and out of the micropores. R: yes, the referee is right in that we are not looking at production and consumption separately; but we only claim the control is on emissions (not production and/or consumption) and we are controlling moisture at the micropore scale. Introduction Lines 210 and 211: concentration) for 24 h, or until the system and the soils atmosphere were emitting low background levels of both N2 and N2O (N2 can get down to levels of 280 ppm much smaller than atmospheric values). Please indicate these "background" values. R: the flushing goes on until there is no further decrease in the background signal. This normally occurs within 24 hours. Values can reach a few gN/ha/d (much lower than atmospheric values of 70%). Lines 222 and 223: Flushing was carried out with He for half an hour before the solution was required for application to the soil cores and continued during the application process to avoid atmospheric N2 contamination (a total of one and a half hours). How this affects the oxygen availability? R: the flushing is done to the amendment outside the incubation vessel, so we remove N2 from the liquid before application. The incubation vessel on the other hand continues to receive He/O2 so it should not affect O2 availability, in fact the increase in CO2 in later experiments supports this assumption. Lines 304 and 305: We accepted these as unavoidable features of the experimental set-up, but we suggest that the main response of the gaseous emissions occurred under the initial conditions, prior to the loss of water over subsequent days. "We suggest" is a strange formulation, either the time coarse of the emissions clearly shows this, or it is an assumption. R: this statement came after a comment from a previous reviewer. We have changed the text now to say 'we assume'. Results Lines 311 UNSAT/halfsat (50-100 N kg- dry soil) Unit of NO3- seems incorrect. Also, the header of Table 2 is wrong (twice UNSAT/SAT)

R: the referee is correct, units and heading have been amended. Lines 349 to 351: The results showed that the total N emission (N2O+N2) (Table 3) had a consistent decreasing trend, with decreasing soil moisture i.e. from 63.4 for SAT/sat (100% WFPS) to 34.1 kg N ha-1 (71% WFPS) for UNSAT/halfsat. I don't see a consistent decreasing trend. Only the driest treatment shows a lower emission. R: we have modified the text to reflect this properly: 'The results showed that the total N emission (N2O+N2) (Table 3) decreased between the highest and the lowest soil moistures i.e. from 63.4 for SAT/sat (100% WFPS) to 34.1 kg N ha-1 (71% WFPS) for UNSAT/halfsat' It also would make more sense to use the same reference for the mineral N content as well as the cumulative gaseous emissions (e.g. per g soil). R: we agree this is a good suggestion. So we have included this extra information in table 3. Lines 351 and 352: The maximum cumulative N2O occurred at around 80% WFPS as Fig. 2 shows. This is an overinterpretation. There are four values and a fit with three unknown is applied. R: we agree that there are no many points, but the value of this analysis is that for a narrow soil moisture range (70-100%) there seems to be a linear response for the N2 but not for the N2O and the total flux. Those shown were the best fits. Noticeable emissions of N2O and N2 occur in all four treatment only up to day four. Bacterial denitrification is identified as the main production pathway. This is due to the experimental setup with a combined amendment of KNO3 and glucose, a setup that produce good conditions for denitrification irrespective of the specific treatment. R: as mentioned earlier, we optimised conditions for denitrification, except for soil moisture that is the factor we are studying.

---

## Author Comment (AC3) · 27 Apr 2017

General remarks This paper presents results from a sophisticated laboratory experiment in which an agricultural soil was compacted and adjusted to 4 different moisture conditions. Glucose and nitrate was added and the formation, isotopic and isoptomeric composition of gasesous N was measured over a period of 12 days. Using those data the authors try to determine the contribution of different processes to N gas formation. The paper is a good example how much information you can get from experimental data if you spend a lot of energy in calculations and data analysis. However, in my eyes the paper has three critical weaknesses:

1.) The results are not really new. It is known for a long time that addition of nitrate and glucose stimulates denitrification in soils and that denitrification is favored under wetterconditions. All the points in the conclusions are not new. If there is new knowledge obtained from the study, it has to be elaborated more clearly. R: we agree that some of the general points are known, for example the effect of soil moisture on emissions, but this is normally considered in relation to ranges of <60%, 60-75% and >75%. We have looked at a more detailed moisture adjustment, four levels at a relatively high moisture range, between 70 to 100% WFPS. We have also studied the isotopocules of N2O and found isotopic similarities at similar moisture levels. Moreover, for the first time we have conducted N2 +N2O flux measurements at defined saturation of pores size fractions as a prerequisite to model denitrification as a function of water status.

2.) The paper is lacking a clear story. It is not really clear to me what was the final purpose of all those detailed analysis. There are some hypothesis mentioned at the end of the introduction but the rest of the manuscript is not tailored to address those hypotheses. The hypothesis that wetter conditions reduce heterogeneity could be answered from just looking at the error bars in figure 1 – you do not need sophisticated analysis to prove this point. Aiming to understand what is going on in one0s own experiment (as stated in the last sentence of the introduction) is not a sufficient aim of a paper. R: We have done a detailed control of soil moisture in the soil and in order to do this we had to do the detailed analysis the reviewer refers to in terms of the moisture adjustment. In this way we ensured that the four moisture levels above 70% WFPS were as accurate as possible. We also used tools such as the isotopomers to confirm source processes, and this is the result of our research in the last 15 years, when we have built up a large database of isotopomers of N2O to improve the uncertainty in the determination of the sources. In this particular experiment we have been able to elucidate the effect of saturation on processes at relatively high moisture levels when combined with the measurements of N2O and N2 emissions.

3.) There are some problems with the experimental approach which limit interpretation of the data. First, moisture conditions were not constant but changed a lot during the experiment. The second treatment, for example at the end of the experiment had

the same water content as the third treatment in the beginning. They had changing substrate concentrations in parallel to changing moisture conditions. Thus, the interpretation of moisture effects during the course of the experiment is difficult. A way to minimize that effect would have been to moisten the supplied He/O2 gas. I would also expect that water loss was highest in the beginning, when the surface layer was drying. A way to get some information about temporal changes of water content would have been to weigh the incubation vessels during the incubation. Second, they measured gas emission – not gas production. They mention this problem in the paper but somehow ignore its consequences. The emitted gas probably originates from those sites which are physically linked to the atmosphere, while gas production, e.g. in the center of aggregates did probably contribute less to the emitted gas. So, the conclusions drawn from the analysis could be valid only for a part of the soil volume. R: we are aware there are limitations to the experimental approach. In order to moist the gas we would have to have an extra vessel where we flush the gas through. Measuring N2 is very difficult due to background atmospheric levels and any additions to the experimental system poses a risk of leaks. In addition, adding moist gas will likely block the tubing as these are very narrow (1/8"o.d.). The flow of the gas is very slow (10 ml/min) simulating a low wind speed so normally this would dry the soil in field conditions too. It would represent a rainfall event where the initial moisture differs between treatments but some drying occurs due to the wind flow. We believe the effect of drying will be more relevant (and significant relative to the initial moisture) later in the incubation. We also know that if drying is significantly affecting the microbes, we would see an increase in CO2 emissions which did not happen later in the incubation. We have introduced changes in the text to make the reader aware of this and have reflected this as 'the effect of initial soil moisture'.

Detailed comments

l.17: remove "soils" R: removed

l.40: What do you mean with "benign" for the environment. Do you mean the process

is important because it closes the global N cycle because it reverses N-fixation? R: no, it is benign because it does not cause harm to the environment.

l.64-73: I would move this paragraph to an earlier point, before talking about compactation. R: we have placed this paragraph after the compaction, as it follows from the previous paragraph where we discuss the effect of livestock on compaction. It also leads to the following text on effect of compaction on soil water: 'reducing the soil air volume and therefore increasing the WFPS'.

l.72: I would replace "powerful tool" by "basis". R: changed

l.81: If there are several references for one statement, present them in chronological order. R: changed

l.81-82: Remove sentence R: removed

l.83: ": : :under the conditions: : :" R: changed

l.92: Be more specific. What do you mean by "other steps of denitrification"? R: we agree that this sentence was not clear enough so we rewrote to: "Simultaneous occurrence production and reduction of N2O as in natural conditions presents a challenge for isotopic factors determination due to uncertainty on N2 reduction and the co-existence of different microbial communities producing N2O (Lewicka-Szczebak et al., 2014). l.93: "reported here". R: changed

l.100: Does that mean that those results are only relevant at elevated C and N? R: We have modified the text as follows: 'The results generally confirmed the range of values of $\eta$ (net isotope effects) and $\eta18O/\eta15N$ ratios reported by previous studies for N2O reduction for that part of the soil volume were denitrification was enhanced by the N+C amendment. This did not apply for the other part of the soil volume not reached by the N+C amendment, showing that the validity of published net isotope effects for soil conditions with low denitrification activity still needs to be evaluated'.

l.108: Why CO2? R: we have changed the text: 'soil to assess the impact of different

levels of soil saturation on N2O and N2 emissions after compaction. CO2 emissions were measured in addition as an estimate of respiration and thus of O2 consumption'. l.112: "controlled laboratory conditions" R: changed but this text is now in section 2.4 as recommended by another referee.

l.119: What do you mean by "heterogeneity in N emissions"? R: spatial distribution of emissions, text changed to clarify

l.120: I am not a soil scientist, but is that really new? R: prediction of N2O emissions is very difficult in part due to their spatial variability. We are trying to understand how this effect occurs in a relatively narrow range of moisture (70-100%). As far as we know there no other studies going to this level of detail. This has been included in the text (end of introduction section).

l.121: Aiming to understand what is going on in one0s own experiment is not a sufficient aim of a paper. R: we have changed the text: 'We aimed to understand changes in the ratio N2O/(N2O+N2) at the different moisture levels studied in a controlled manner on soil micro and macropores. Moreover, we used isotopocule values of N2O to evaluate if the contribution of bacterial denitrification to the total N2O flux was affected by moisture status'

l.137: Verb missing. "was applied"? R: the verb is early on in the paragraph. The paragraph is now split to make it clear.

l.228: "CO2 was measured: : :" R: changed

l.230: replace "pulled together in one sample" by "pooled" R: changed

l.232: Remove sentence. There is a similar sentence in the results section. R: removed

l.268: Were the data normal distributed? R: yes, all datasets were tested by fitting a Gaussian model resulting in Fprob<0.001. this was added in the results section.

l.275: "mixing model was then used" (use past tense) R: changed

l.283: When did this occur and what is a possible explanation? Wrong fractionation factors? We clarified the variability of endmember values and fractionation factors in the introduction: "The analysis comprised measurements of the N2O and N2 fluxes combined with isotopocule data. Net isotope effects ($\eta$ values) are variable to a certain extent as they result from a combination of several processes causing isotopic fractionation (Well et al., 2012). The results generally confirmed the range of of $\eta$ values and $\eta$18O/$\eta$15N ratios reported by previous studies for N2O reduction for the soil volume reached by the N+C amendment. This did not apply for the soil volume not reached by the N+C amendment."

l.290: A TCD is an detector – not an analyzer. R: changed analysed for determined

l.303 Why was the gas stream not bubbled through water to saturate it with water? R: see our explanation above in point 3.

l.305: I would expect the highest water loss right in the beginning. R: the flowrate is very low so drying will take a while, we are assuming that the significant water loss will affect later in the incubation, later than the peaks appear. However, as explained earlier, we have now referred to the effect of the initial soil moisture in the treatments.

l.306. But they were similar between treatments in the end although different starting conditions. R: yes

l.314-316: There was a high variability in the data. R: but only for NH4+ it was not significant. A sentence was added

l.318: Remove "The results showed that" R: removed

l.329: I do not see that in Figure 1. In Unsat/sat the N2O maximum was at 12 kg N/ha d, not around 7. R: the referee is correct, we have now amended the text to reflect this: 'The N2O maximum in the SAT/sat and HALFSAT/sat treatments was of similar magnitude (means of 5.5 and 6.5 kg N ha-1 d-1, respectively) and but not those of UNSAT/sat and UNSAT/halfsat (means of 7.1 and 11.9 kg N ha-1 d-1, respectively).

l.348. Right. But what are the consequences of this for your experiment and its interpretation? R: this belongs to the discussion (4.1) so have been moved in there to explain the potential underestimation of the production due to low diffusion.

l.354: You probably mean "CO2 fluxes". Why was CO2 measured? R: yes, added fluxes in the sentence. CO2 indicates aerobic respiration and as explained above (l.108) is also affected by the soil moisture and level of compaction.

l.360: The carbon budget is interesting but complicated. Could you calculate recovery rates for the added glucose? It looks as if there are recoveries higher than 100%. Can this be interpreted as a priming effect? A problem with using CO2 for carbon budgeting is, that depending on pH you also have other IC species in the soil solution. Do you know the pH in your soils? R: pH is 5.63 as shown in Table 1. We did not do a C budget, but it is possible that soil C would have also contributed to the CO2 emitted but to a lower extent compared to the added glucose.

l.370: Add article before "period" R: added

l.375: The SP data have a high standard deviation. Are the differencers discussed in this paragraph real? R: we think the larger variation (high SD) of SP around day 3 corresponds to the with highest variation of N2 and N2O fluxes (which is evident from Figs

l.391:You may consider adding these data to the plot. R: data added to figure

l.394: Separate into two sentences. Start second one with "In our data, maximum : : :." R: changed

l.404 So what is the message ofthis paragraph with respect to the first sentence of the paragraph? R: we have rewritten: 'the question arises to which extent the relationships between the d18O and d15Nbulk and between d18O and SP within the individual treatments denitrification dynamics. We checked this to evaluate the robustness of isotope effects during N2O reduction as a prerequisite to calculate the percentage of bacterial

denitrification in N2O production."

l.405: Why was this done? R: we have found that the isotopologues seem to be potentially more powerful than initially thought. By looking at these relationships we have learnt how the responses relate to the sources of these gases.

l.428: Why was this plot done? R: the same reason as above

l.441: I do not see data within those areas in the plots. R: we have not been so clear, and we refer to the vectors more than the areas. Text has been changed to reflect this.

l.456: "sat" page 19: It is difficult to detect the storyline on this page. R: we are explaining that from our results we are providing a refinement in the soil moisture (WFPS) thresholds previously established as borderline for nitrification-denitrification. We are also proposing that WFPS which was previously established as a normalised parameter for these type of soil moisture thresholds, might actually change with soil type.

L513: Could it be that there was C limitation in the dryer treatments because glucose was metabolized aerobically? R: if glucose was metabolised we would have expected C to have been less limiting

l.534-537: The message of the CO2 paragraph is not really clear. Are the CO2 data helpful in this manuscript? R: we have deleted the paragraph as suggested.

l.539: How much is the unaccounted N-loss in comparison to the accounted gasesous losses? R: we added: " unaccounted-for N loss is two to three times the total measured gas loss (Table 3)".

l.541: NO: What are typical NO fluxes in the literature? Can the NO flux have a significant magnitude? The same applies to microbial biomass: Is the microbial biomass potentially formed from the unaccounted N-loss in a realistic order of magnitude? R: we are now able to measure NO fluxes in the system. Loick et al reports a ratio N2O/NO of 0.4 for example, so yes, it can be significant. We did not do microbial biomass in this instance.

l.567: How should nitrification contribute to BDEN? Do you mean nitrifier-denitrification? R: thus large contributions to the total N2O flux from nitrification were not probable

l.636: I do not understand the content and purpose of this paragraph. R: text changed to: The question arises, if the poor coincidence of Pool 2 isotopologue fluxes with previous N2O reduction studies reflects the variability of isotope effects of N2O reduction or if the contribution of other processes like fungal denitrification could explain this (Lewicka-Szczabek et al, 2017). The latter explanation is evaluated in section 4.3.

l.719: Don0t you have 4 periods in the figure? Table 3: Unit missing for Total emitted N. Tables 5 and 6: I wonder whether these data could be presented better in figures. R: no, only three. Units included. Yes, figures can illustrate better, but as we explained in the initial review, this data is very useful for models and we think providing the values will be more useful.

Figure 5: the four sub-graphs are quite similar. Isn0t a conclusion that the results were not much influenced by soil moisture? Do you really need 4 graphs? R: we concluded that there were similarities between the 2 high moisture and 2 low moisture treatments. We believe this is an important finding due to the relatively narrow range of soil moisture we have studied, above 70%, in which we still find differences in fluxes. Davidson stated that the threshold for nitrification-denitrification lies at about 60%, in our case we have managed to refine this.

---

## Referee Report (RR1)

**Evaluation of manuscript Cardenas et al. Effect of soil saturation on denitrification in a grassland soil (Bg-2016-556)**

Authors have improved the manuscript after this revision, solving most of the weak points of the initial MS. Now the arguments done in the MS are easier to follow by readers. The paper has novelty and interest from a mechanistic point of view, especially due to the information obtained about the isotopic signal of N2O emitted under denitrification conditions. Results confirmed that there were 2 different N pools in soil with different dynamic. Therefore, I recommend its publication in the current form.

---

## Author Response (AR3)

**Referee 1**

Authors have improved the manuscript after this revision, solving most of the weak points of the initial MS. Now the arguments done in the MS are easier to follow by readers. The paper has novelty and interest from a mechanistic point of view, especially due to the information obtained about the isotopic signal of N2O emitted under denitrification conditions. Results confirmed that there were 2 different N pools in soil with different dynamic. Therefore, I recommend its publication in the current form.

R: We thank the reviewer for the comments that made the manuscript greatly improved.

**Referee 2:**

Nitrous oxide is a powerful greenhouse gas, the pathway of N2O production is important to model and mitigate N2O emissions, which would be changed due to differ in moisture contents. The authors seek to evaluate the roles of soil water filled pore space and nitrous oxide emission and its pathway. They established a laboratory incubation experiment in a relatively narrow range of moisture (70-100%), emissions of N2O, N2 and CO2, as well as the isotopocules of N2O, were measured. The authors found that denitrification as the main source of fluxes at the highest saturations, but nitrification could have occurred at the lower saturation, even though moisture was still high (71% WFSP). The results indicated that there are 2 N-pools with different dynamics: added N producing intense denitrification, vs soil N resulting in less isotopic fractionation.
Lots of studies have reported the relationship between the moisture contents and denitrification and N2O and N2 emission, and the effects of spatial heterogeneity and source processes on GHG. Whilst, the study conducted in a relatively narrow range of moisture is rare. These results could contribute to model and mitigate GHG emissions from agricultural soils. Additionally, the authors have revised the paper according the first 3 reviewers, which improved the paper a lot. In my opinion, this is an interesting paper, although it has some small problems should be revised. Thus, it can be accepted after minor revision. My suggestions were shown as follows.

Line 121 and 22, the authors hypothesized that, even at high soil moisture, a mixture of nitrification and denitrification can occur. However, the authors haven't explain the base of this hypothesis.

R: we base this hypothesis on the fact that pockets of aerobicity as well of anaerobicity can occur at high soil moisture, mainly driven by soil respiration after application of N and C (using up O2) and further recovery after nutrients are used becoming limiting (increasing aeration). Text has been added to explain this.

Line 183, the incubation experiment lasted 13 days. The detail of incubation term should be described, "one day before amendment application and 12 days after ….". Otherwise, the readers would be puzzled.
R: text amended as recommended

Line 268, in "Results" section, the results were described too much in details. In my view, the results which supported the conclusions could be described in details, whist, the other results may be described in brief. In fact, it is difficult for the reads to remember so much information and to find the important information for the conclusions.
R: most of the details provided are the result of recommendations of previous reviews (5 in total before these three). For example:

**Referee Lines 349 to 351: The results showed that the total N emission (N2O+N2) (Table 3) had a consistent decreasing trend, with decreasing soil moisture i.e. from 63.4 for SAT/sat (100% WFPS) to 34.1 kg N ha-1 (71%**

**WFPS) for UNSAT/halfsat. I don't see a consistent decreasing trend. Only the driest treatment shows a lower emission.**
Response: we have modified the text to reflect this properly: 'The results showed that the total N emission (N2O+N2) (Table 3) decreased between the highest and the lowest soil moistures i.e. from 63.4 for SAT/sat (100% WFPS) to 34.1 kg N ha-1 (71% WFPS) for UNSAT/halfsat'

Also:

**Referee: Line 367-369: The writing is confusing here; I cannot follow the meaning of this sentence.'**
Response: These are the lines in the submitted pdf: "This is in juxtaposition with the situation when the N emissions are dominated by N2 or N2O is low, where the SP values of soil emitted N2O were much higher (Fig. 3), pointing to an overall product ratio related to an 'isotopic shift' of 10 to 12.5o/oo."
We modified to (including previous sentence):
"The plot of the N2O / (N2O + N2) ratio vs SP for all treatments in the first two days (when N2O was increasing and the N2O / (N2O + N2) ratio decreasing) shows a significant negative response of the SP when the ratio increased (Fig. 3). The regression suggests that when the emitted gaseous N is dominated by N2O (ratio close to 1) the SP values will be slightly negative with values around -2 (Fig. 3), i.e. within the range SP range of bacterial denitrification. With decreasing N2O / (N2O + N2) ratio the SP values of soil emitted N2O were increasing to values up to 8 per mil."

Line 364 to 368, in "Results" just showing the results of the experiment. Thus, those sentences may be moved to "Discussion" section.
R: sentences have been moved as recommended

Line 385 to 389, move those sentences to Fig. 4 as a footnote.
R: text has been moved

Line 402, "2-pool dynamics" should be described.
R: this reflects the competition for soil N vs applied N as described in Lewicka-Szczebak et al. (2015). Reference is already mentioned in the paragraph, and we had said: 'Pool 1 representing amended soil….and added 'whereas for Pool 2 (amendment-free soil)… '

Line 434, the topic of different paragraphs might be used as the subtitle in "Discussion" section, which might be easy for readers to understand and to follow authors' thought. For example, in 4.1, "Nitrogen emission increasing with moisture contents" may be used as the subtitle of this section.
R: sub headings were added to help clarity

Fig. 1, take the figure of N2O, N2 or CO2 together, and then they would be easily compared.
R: we have joined N2O and N2, left CO2 separate as the figure becomes too busy and difficult to see.

Fig. 4, the icon of Fig. 4 should be described.
R: we have added text in the legend to describe the periods. We are not sure if the reviewer means something else.

Fig. 5, "BD", "ND" and "FD" should be described.

R: these are already explained in the legend: 'Endmember areas for nitrification, N; bacterial denitrification, D; fungal denitrification, FD and nitrifier denitrification, ND'.

**Referee 3**

The authors should emphasise the novelty of the combination of approaches used, i.e.

i) that the ms addresses the relationship between %WFPS and DIRECT measurement of both N2O and N2. (It does not solely focus on the well documented relationship between N2O fluxes and WFPS), nor does it use the acetylene inhibition technique which has its own deficiencies - especially in fine textured soils and high soil moistre contents,

R: text re. moisture and acetylene was added in the introduction: 'They mostly rely on the knowledge of the effect of moisture on soil processes, whilst in our study, we combined direct measurements of both N2O and N2 with isotopomers of N2O to verify the source processes.' '. The

N2 emissions were based on direct measurements from the incubated soils, avoiding methodologies that rely on inhibitors such as acetylene with limitations in diffusion in soil and causing the oxidation of NO (Nadeem et al., 2013).'

ii) it combines the measurement approach with N2O isotopomer measurements to verify both the pool of N emitted as N2O, and verify the dominant soil microbial processes responsible for the N2O and N2 fluxes, and

R: text has been added in the conclusions: 'This study combined direct measurements of N2 as indicator of denitrification with isotopomers providing a measurement approach that verifies the source processes of N2O emissions.'

ii) it uses a precise means of compacting the soil and generating the initial %WFPS treatments.

R: text was added in the introduction to enhance the importance of this: 'In addition, the packing of the cores in our study was of great precision increasing our potential to achieve reproducibility in the replicates where a mixture of aerobic/anaerobic pores might have occurred. '

The authors should also expand the discussion to address the assumptions made about the distribution of the added glucose and NO3 within the soil profiles of the different treatments and subsequent intepretation of the results. I do not see this as a reason to prevent publication - but I do think that the authors could comment on this in the discussion a little more. It's a shame that this could not have been verified in a parallel set of soil cores using labelled C and N. Can the authors also comment on whether there was any leaching/loss of the glucose and nitrate additions from the soil 'cores'.

R: thanks to the reviewer for these useful comments.

We were not able to simultaneous labelling and emissions measurements of $N_2O$ and $N_2$ at the time of this study, in fact we are not aware of any study which have ever done this before, but this is now a facility we have developed. We have however in the recent past used labelling techniques to help us determine the contribution of the added pool to the emissions. For example in Loick et al. (2016) we estimated that the proportion of N2O emitted that derived from the added pool of N and C was ca. 85% under denitrifying conditions. However, this does not give us direct information of the amendment distribution. Also in Loick et al. (2016), we undertook preliminary tests to determine infiltration of the amendment using dyes, but it was difficult to be certain on where the amendment went.

However, in previous studies we assumed (for modelling purposes) multiple pools after N and glucose amendment. In Bergstermann et al. (2011) for example we presumed they occupied 10% of the core volume (pool 1), because this resulted in a good fit for measured and modelled $N_2$ and $N_2O$ fluxes as well as $d^{15}N^{bulk}$ values. In the current study, we could assume that in the wettest treatment this (proportional) volume was smaller ie similar to the pore volume displaced by the added 5 ml of amendment since pores were almost completely filled with water. Furthermore, that it would have been the largest in the driest treatment where the amendment solution was able to infiltrate the partly saturated pore space and thereby increasing the water content in the infiltrated volume. With regards to leaching, it was minimal (< 0.5 mL water in the core) and so significant leaching of amendment can thus be excluded.

Text has been added to the manuscript in the discussion.